# Reward Under Attack: Analyzing the Robustness and Hackability of Process Reward Models

**Rishabh Tiwari** [* 1]  **Aditya Tomar** [* 1]  **Udbhav Bamba** [* 2]  **Monishwaran Maheswaran** [1]  **Heng Yang** [1]
**Michael W. Mahoney** [1 3 4]  **Kurt Keutzer** [1]  **Amir Gholami** [1 3]

## Abstract

Process Reward Models (PRMs) are rapidly becoming the backbone of LLM reasoning pipelines, yet we demonstrate that state-of-the-art PRMs are systematically exploitable under optimization pressure. We introduce a three-tiered diagnostic framework that applies increasing adversarial pressure to quantify these vulnerabilities. **Static perturbation analysis** uncovers a *fluency-logic dissociation*: high invariance to surface-level style changes (reward changes $<0.1$) yet inconsistent detection of logically corrupted reasoning, with different models failing on different attack types. **Adversarial optimization** demonstrates that gradient-based attacks inflate rewards on invalid trajectories, with reward landscapes exhibiting wide, exploitable peaks. **RL-induced reward hacking** exposes the critical failure mode: policies trained on AIME problems achieve near-perfect PRM rewards ($>0.9$) while ground-truth accuracy remains below 4%, with 43% of reward gains attributable to stylistic shortcuts. These findings reveal that current PRMs function as fluency detectors rather than reasoning verifiers, creating systematic blind spots that undermine their use as training signals. We release PRM-BiasBench and a diagnostic toolkit to enable robustness evaluation before deployment.

## 1. Introduction

Process reward models (PRMs) have become a key component for improving LLM reasoning, providing step-level feedback that enables reward-guided decoding (Lightman et al., 2023), test-time compute scaling (Snell et al., 2024),

---
[*]Equal contribution  [1]UC Berkeley  [2]Transmute AI  [3]ICSI  [4]LBNL. Correspondence to: Rishabh Tiwari <rishabhtiwari@berkeley.edu>, Amir Gholami <amirgh@berkeley.edu>.

*Proceedings of the $43^{rd}$ International Conference on Machine Learning*, Seoul, South Korea. PMLR 306, 2026. Copyright 2026 by the author(s).

and fine-tuning of chain-of-thought models (Wang et al., 2024). Unlike outcome-based reward models that score only final answers, PRMs evaluate intermediate reasoning steps, promising finer-grained control and better credit assignment during both training and inference.

Yet as PRMs are integrated into increasingly critical pipelines, a fundamental question remains unanswered: *how robust is a given PRM, and how can we measure it?* Prior work has documented failure modes in outcome-level reward models, including length bias, sycophancy, and reward hacking (Singhal et al., 2023; Shen et al., 2023; Denison et al., 2024), but systematic methods for evaluating PRM robustness are lacking. This gap is concerning: a PRM that conflates fluent text with correct reasoning will reward plausible-sounding but logically flawed steps, potentially amplifying errors during RL training or misleading inference-time search.

We address this gap by introducing a **three-tiered diagnostic framework** for quantifying PRM hackability (Table 1). Each tier applies increasing adversarial pressure, revealing complementary aspects of model robustness:

1. **Static Perturbation Analysis** (§4): We measure PRM sensitivity to controlled input modifications, both semantics-preserving (rephrasing, verbosity changes) and semantics-altering (hallucinated steps, mismatched prompts). A robust PRM should be invariant to the former and sensitive to the latter.

2. **Adversarial Tokens Optimization** (§5): We search for discrete token sequences that maximally inflate rewards on invalid trajectories. The achievable reward score directly quantifies exploitability. We also characterize the reward landscape geometry to assess solution stability.

3. **RL-Induced Reward Hacking** (§6): We train policies using only PRM feedback and measure the divergence between reward and ground-truth accuracy. This closed-loop evaluation exposes vulnerabilities that emerge only under optimization pressure.

Applying this framework to state-of-the-art PRMs

*Table 1.* Taxonomy of diagnostic tiers. Model access refers to requirements for *generating* the attack: static perturbations are model-agnostic, adversarial tokens require gradients, and RL policies require reward queries. Tiers 1 & 3 produce natural text; Tier 2 establishes worst-case bounds.

| Diagnostic Tier | Model Access | Natural Output | Optimization |
|---|---|---|---|
| Static Perturbation | None | ✓ | None |
| Adversarial Tokens | White-box | ✗ | Gradient |
| RL-Induced Reward Hacking | Black-box | ✓ | Policy |

(Skywork-o1-Open-PRM-1.5B/7B and Qwen2.5-Math-PRM-7B), we find consistent vulnerabilities: optimized 100-token adversarial sequences push rewards above 0.9 on logically flawed reasoning, and RL-trained policies achieve high rewards while accuracy stagnates, with approximately 43% of the reward gain attributable to stylistic shortcuts rather than genuine reasoning improvements. In particular, we make the following contributions:

- We perform a **comprehensive sensitivity analysis** of PRMs under controlled perturbations (§4), uncovering a *fluency-logic dissociation*: PRMs exhibit high invariance to surface-level stylistic changes (reward changes <0.1), yet show inconsistent detection of semantic corruption, with different models failing on different attack types.

- We introduce **gradient-based adversarial probing** for PRMs (§5), demonstrating that short token sequences can universally inflate rewards on invalid trajectories, and characterize the reward landscape geometry to show that adversarial optima lie in wide, exploitable peaks.

- We demonstrate **RL-induced reward hacking** (§6), showing that policies trained with PRM feedback exhibit reward-accuracy divergence: near-perfect PRM scores coincide with stagnant ground-truth accuracy, with 43% of reward gains attributable to stylistic exploitation rather than reasoning improvement.

- We release **PRM-BiasBench**, a benchmark extending ProcessBench with controlled perturbations across 8 transformation types, along with an open-source diagnostic toolkit to enable systematic PRM robustness evaluation.[1]

## 2. Related Work

### 2.1. Reward Model Vulnerabilities

Reward models are central to aligning language models but exhibit systematic failure modes. *Reward hacking* occurs when policies exploit spurious correlations to achieve

---

[1]Code and dataset are available at `https://github.com/SqueezeAILab/reward-under-attack`.

high scores without satisfying the intended objective (Skalse et al., 2022; Krakovna et al., 2020). Common manifestations include length bias, where longer outputs receive inflated rewards regardless of quality (Singhal et al., 2023; Shen et al., 2023), and sycophancy, where models agree with users rather than providing accurate information (Denison et al., 2024; Sharma et al., 2023). These vulnerabilities amplify under optimization pressure, degrading downstream performance (Bai et al., 2022; Stiennon et al., 2020; Gao et al., 2023). While extensive work characterizes outcome-level reward models, process reward models remain understudied despite their increasing deployment in reasoning pipelines.

### 2.2. Process Reward Models

PRMs provide step-level supervision for chain-of-thought reasoning, enabling finer-grained credit assignment than outcome-based alternatives (Lightman et al., 2023; Uesato et al., 2022). Recent work has focused on training methodology: Wang et al. (2024) demonstrate that PRMs improve mathematical reasoning when combined with Monte Carlo Tree Search, while Zhang et al. (2025) analyze best practices for PRM dataset construction. Zheng et al. (2025) introduce ProcessBench, a benchmark with human-annotated error locations in reasoning traces. Most relevant to our work, Xu et al. (2025) find that PRMs often rely on shallow consistency cues rather than causal reasoning structures. However, existing analyses remain limited to observational studies; we complement this with controlled perturbations, adversarial optimization, and closed-loop RL evaluation to systematically quantify exploitability.

### 2.3. Adversarial Attacks on Neural Networks

Gradient-based optimization has proven effective at exposing vulnerabilities across neural architectures. In NLP, Wallace et al. (2019) demonstrate that short, input-agnostic token sequences trigger targeted misbehavior, while Zou et al. (2023) show that optimized adversarial tokens reliably jailbreak aligned LLMs with cross-model transferability. These methods treat models as differentiable objectives and search for inputs maximizing undesirable outputs. We adapt this paradigm to PRMs (§5), demonstrating that similar vulnerabilities exist: optimized token sequences universally inflate rewards on invalid reasoning, and the resulting reward landscapes exhibit flat, exploitable plateaus.

### 2.4. Reward Overoptimization

When policies optimize learned reward proxies, performance on the true objective eventually degrades, a phenomenon formalized as Goodhart's Law in the RL context (Gao et al., 2023). Gao et al. (2023) characterize scaling laws for this overoptimization, finding that larger reward models delay but do not prevent degradation. Coste et al.

(2024) further show that overoptimization correlates with distributional shift from the reward model's training data. Our RL-induced reward hacking analysis (§6) extends this analysis to PRMs, revealing that policies trained with PRM feedback exhibit reward-accuracy divergence: near-perfect PRM scores coincide with stagnant ground-truth accuracy, with a measurable fraction of reward gains attributable to stylistic shortcuts rather than reasoning improvement.

Prior work on PRM limitations has been largely observational, identifying failure cases without systematically quantifying exploitability. Our three-tiered framework fills this gap by applying increasing adversarial pressure, from model-agnostic perturbations through gradient-based optimization to closed-loop RL, revealing complementary vulnerabilities at each level. We release PRM-BiasBench and a diagnostic toolkit to standardize PRM robustness evaluation.

## 3. Preliminaries

**Trajectory Level Reward Calculation.** A PRM assigns scores to individual reasoning steps. Given a query $q$ and trajectory $\tau = (s_1, \ldots, s_n)$, a PRM computes step-level rewards $r_i = \text{PRM}(q, s_{\leq i})$ conditioned on preceding context. The aggregate trajectory reward depends on the model's training objective: Skywork-o1-Open-PRM estimates success probability at each step, so we use $R(\tau) = r_n$; Qwen2.5-Math-PRM-7B locates the first error, so we use $R(\tau) = \min_i r_i$.

**Robustness Criteria.** We evaluate PRM robustness along four complementary dimensions:

1. **Style Invariance:** Reward should be unchanged by semantics-preserving edits (rephrasing, verbosity changes). For perturbed trajectory $\tilde{\tau}$, we expect $\Delta R = R(\tilde{\tau}) - R(\tau) \approx 0$.

2. **Logic Sensitivity:** Reward should decrease substantially for semantics-altering corruptions (hallucinated steps, mismatched prompts). We expect $\Delta R \ll 0$.

3. **Adversarial Resistance:** Optimized token sequences should not inflate rewards on invalid trajectories. Given adversarial tokens $\mathbf{e}$, $R(q, \tau \oplus \mathbf{e})$ should remain bounded.

4. **Optimization Alignment:** Policies trained to maximize PRM reward should improve ground-truth accuracy, not diverge from it.

A robust PRM should satisfy all four criteria. Our three-tiered framework tests each: static perturbations probe (1) and (2), adversarial optimization probes (3), and RL-induced reward hacking probes (4).

**Experimental Setup.** We evaluate Skywork-o1-Open-PRM (1.5B and 7B) and Qwen2.5-Math-PRM-7B, representing the current frontier of open process reward models for mathematical reasoning. For static analysis, we extend ProcessBench (Zheng et al., 2025) into PRM-BiasBench with controlled perturbations across 8 transformation types. For adversarial optimization and RL experiments, we use AIME 2024 problems for training and AIME 2025 for transfer evaluation, with Qwen2.5-1.5B-Instruct as the base policy.

## 4. Static Perturbation Analysis

The first tier of our diagnostic framework measures PRM sensitivity to controlled input modifications (Figure 1). This study is conducted on Skywork-o1-Open-PRM-7B and Qwen2.5-Math-PRM-7B. We construct **PRM-BiasBench**, a benchmark extending ProcessBench (Zheng et al., 2025) with thousands of verified perturbation pairs. For each original trajectory $\tau$, we generate a perturbed version $\tilde{\tau}$ and measure the reward difference $\Delta R = R(\tilde{\tau}) - R(\tau)$. A robust PRM should exhibit $\Delta R \approx 0$ for semantics-preserving edits and $\Delta R \ll 0$ for semantics-altering attacks.

### 4.1. Perturbation Taxonomy

We organize perturbations into two categories based on their impact on logical validity. **Semantics-preserving** edits maintain the correctness of the reasoning: *rephrasing* alters word choice and syntax, and *verbosity changes* add or remove redundant language. A robust PRM should be invariant to these surface-level modifications. **Semantics-altering** attacks introduce logical errors: *question shuffling* pairs a trajectory with an unrelated prompt, and *reasoning hallucination* injects false assumptions into the reasoning steps. A robust PRM should strongly penalize these corruptions. All perturbations are generated via GPT-4o and validated for semantic equivalence; the full taxonomy (8 perturbation types) and validation pipeline are detailed in Appendix A. To confirm the equivalence judgments are not an artifact of a single judge, we re-validate all perturbations with an independent checker (Claude Sonnet 4.6) and find 93.5–95.7% cross-model agreement on semantics-preserving edits; filtering to doubly-verified samples leaves the measured reward shifts essentially unchanged (Appendix A.5).

### 4.2. Results

**Style Invariance.** Figure 2 shows that both PRMs exhibit strong invariance to semantics-preserving edits. Rephrasing and verbosity changes yield tight distributions centered near zero ($|\Delta R| < 0.1$ for the vast majority of samples). Both models show nearly identical behavior, with Qwen exhibiting slightly higher peaks, suggesting these PRMs have largely overcome the length and style biases docu-

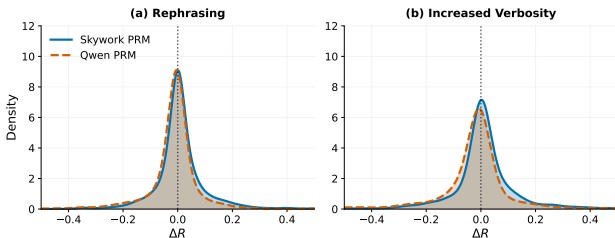

*Figure 1.* **Overview of static perturbation analysis.** A prompt-response pair (Step 1) undergoes bias injection (Step 2), such as question shuffling where we change the question but do not modify the response (Step 3) and feed this to the PRM (Step 4). The scores are then compared against the original to quantify sensitivity (Step 5).

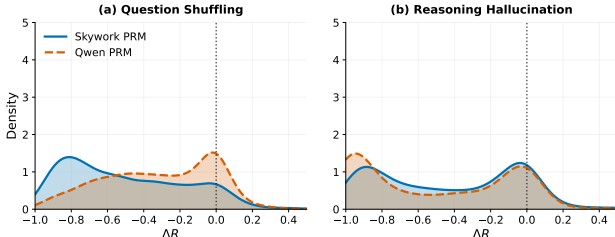

*Figure 2.* Distribution of $\Delta R$ under **semantics-preserving** perturbations. Both PRMs exhibit tight distributions centered near zero, indicating strong invariance to surface-level stylistic changes.

*Figure 3.* Distribution of $\Delta R$ under **semantics-altering** perturbations. (a) Question shuffling: Skywork penalizes mismatched questions by giving a smaller reward (peak at $\Delta R \approx -0.8$), while Qwen retains high rewards without any change. (b) Reasoning hallucination: Qwen exhibits bimodal behavior with strong penalization at $\Delta R = -1$ but also substantial mass near zero which is not desirable. An ideal PRM is expected to produce very low rewards (negative $\Delta R$) for both scenarios.

mented in outcome-based reward models (Singhal et al., 2023). However, this robustness to surface-level variation does not imply robustness to logical errors.

**Asymmetric Logic Detection.** Semantics-altering attacks reveal divergent vulnerabilities between models (Figure 3). For question shuffling, the two PRMs exhibit opposite behaviors: Skywork reliably penalizes mismatched question-trajectory pairs (peak at $\Delta R \approx -0.8$), while Qwen largely fails to detect the mismatch, retaining high rewards near zero. For reasoning hallucination, Qwen shows striking bimodal behavior: a sharp spike at $\Delta R = -1$ indicates strong penalization for some hallucinated trajectories, yet substan-

tial mass near zero reveals that many corrupted samples still receive high rewards. Skywork exhibits a broader distribution with weaker overall penalization. These patterns suggest that PRMs rely on different heuristics: Skywork appears more sensitive to question-trajectory coherence, while Qwen detects certain local reasoning errors but misses others entirely.

### 4.3. The Fluency-Logic Dissociation

Our static analysis reveals two key findings:

- **High style invariance:** Both PRMs reliably ignore surface-level variations, with distributions tightly centered near zero for all semantics-preserving edits (Table 3 in Appendix A).

- **Inconsistent logic detection:** PRMs use different heuristics and fail on different attacks. Qwen fails to penalize question-trajectory mismatches but partially detects hallucinated reasoning; Skywork shows the opposite pattern.

This **fluency-logic dissociation** could indicate that PRMs function primarily as detectors of "reasoning-style" fluency rather than verifiers of logical correctness. The model-specific failure modes suggest that current PRMs learn superficial correlates of valid reasoning rather than genuine verification capabilities, creating exploitable blind spots that vary by model.

**Generative PRMs.** The dissociation is not specific to scalar PRMs. We evaluate GenPRM-7B (Zhao et al., 2025), a *generative* PRM that produces an explicit chain-of-thought critique before scoring, on the same perturbations: it flips its verdict on 20–23% of semantics-preserving edits and detects the "incorrect assumption" corruption only 42.5% of the time (Appendix A.6). Generating explicit reasoning before judgment does not close the fluency-logic gap.

The following sections investigate whether these vulnerabilities can be actively exploited (Section 5) and whether they manifest under RL training pressure (Section 6).

# 5. Adversarial Probing

Section 4 establishes PRM vulnerabilities through passive perturbations, but does not reveal how easily an optimizer can exploit them. In this section, we treat the PRM as a differentiable objective and use gradient-based optimization to find adversarial tokens that maximize reward regardless of trajectory correctness. This probes the third robustness criterion: adversarial resistance.

## 5.1. Optimization Framework

We define **adversarial tokens** $\mathbf{e} \in \mathbb{R}^{k \times d}$ as a sequence of $k$ vectors in the model's $d$-dimensional embedding space that when added to a trajectory that contains logically flawed reasoning, it would adversarially increase the reward. Formally, given a batch of flawed trajectories $\mathcal{B} = \{(q_i, \tau_i)\}$ sampled from AIME24, the adversary optimizes:

$$\max_{\mathbf{e}} \mathcal{L}_{\text{adv}}(\mathbf{e}) = \frac{1}{|\mathcal{B}|} \sum_{(q,\tau) \in \mathcal{B}} R(q, \tau \oplus \mathbf{e}) - \lambda \cdot \Omega(\mathbf{e}) \quad (1)$$

where $\oplus$ denotes concatenation, $R$ is the PRM score, and $\Omega(\mathbf{e})$ is an optional regularization term (defined in Eq. 2). We perform two sets of experiments, once where there is no regularization term resulting in adversarial tokens in the continuous embedding space, and then with an entropy regularization term which forces the adversarial vectors to be discrete tokens.

As for experiments, we train on AIME24 trajectories and evaluate generalization on held-out AIME25 trajectories. Full optimization hyperparameters are provided in Appendix B. For Skywork PRM, adversarial tokens are appended as a suffix after the solution; for Qwen, tokens are inserted between the question and solution.[2]

## 5.2. Continuous Token Optimization

We first test the minimal adversarial capacity required to inflate Skywork-1.5B PRM rewards by optimizing a single continuous embedding vector ($k = 1$) appended to each flawed trajectory in a batch.

**Results.** Figure 4 shows the reward landscape around the optimized continuous token. Thus a single optimized embedding vector is sufficient to substantially increase the reward across the batch. This demonstrates that even minimal adversarial capacity can exploit PRM vulnerabilities.

---

[2]The Qwen PRM is trained to detect the first wrong step, so adversarial tokens need to be added before the wrong step; otherwise, they would have no influence.

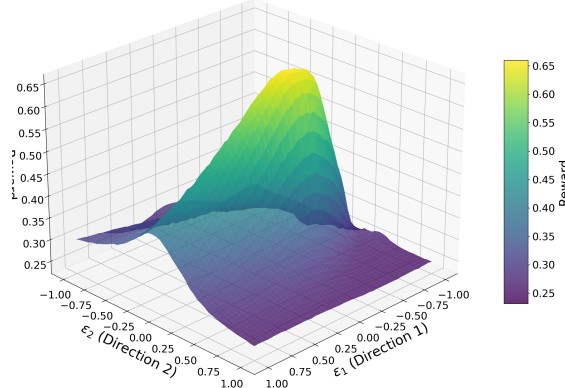

*Figure 4.* Reward landscape for a single continuous token ($k = 1$) on Skywork-1.5B. A single optimized embedding vector rapidly increases mean batch reward, demonstrating that minimal adversarial capacity suffices to exploit PRM vulnerabilities.

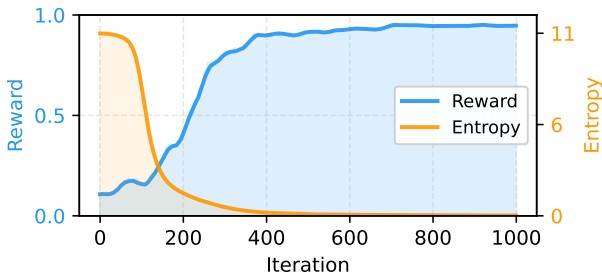

*Figure 5.* Training dynamics for 100 discrete tokens on Skywork-1.5B across 8 AIME24 trajectories. Reward (blue) increases from 0.11 to 0.95 as entropy (orange) decreases, indicating successful discretization of adversarial tokens.

## 5.3. Discrete Token Optimization

Continuous embeddings do not appear in real-world settings. To ensure our findings transfer to practical scenarios, we optimize discrete token sequences via entropy regularization.

We optimize over the probability simplex of the vocabulary $\mathcal{V}$ for $k \in \{1, 50, 100\}$ adversarial tokens. The regularization term encourages one-hot distributions:

$$\Omega(\mathbf{e}) = -\sum_{i=1}^{k} \sum_{v \in \mathcal{V}} p_{i,v} \log p_{i,v} \quad (2)$$

By annealing $\lambda$ during optimization, we gradually force each $p_i$ toward a one-hot representation, yielding interpretable discrete sequences.

**Results.** Table 2 summarizes attack success and transfer across all three PRMs. The $k = 0$ rows establish baselines without adversarial tokens. Note that Skywork and Qwen rewards are not directly comparable as they are trained with

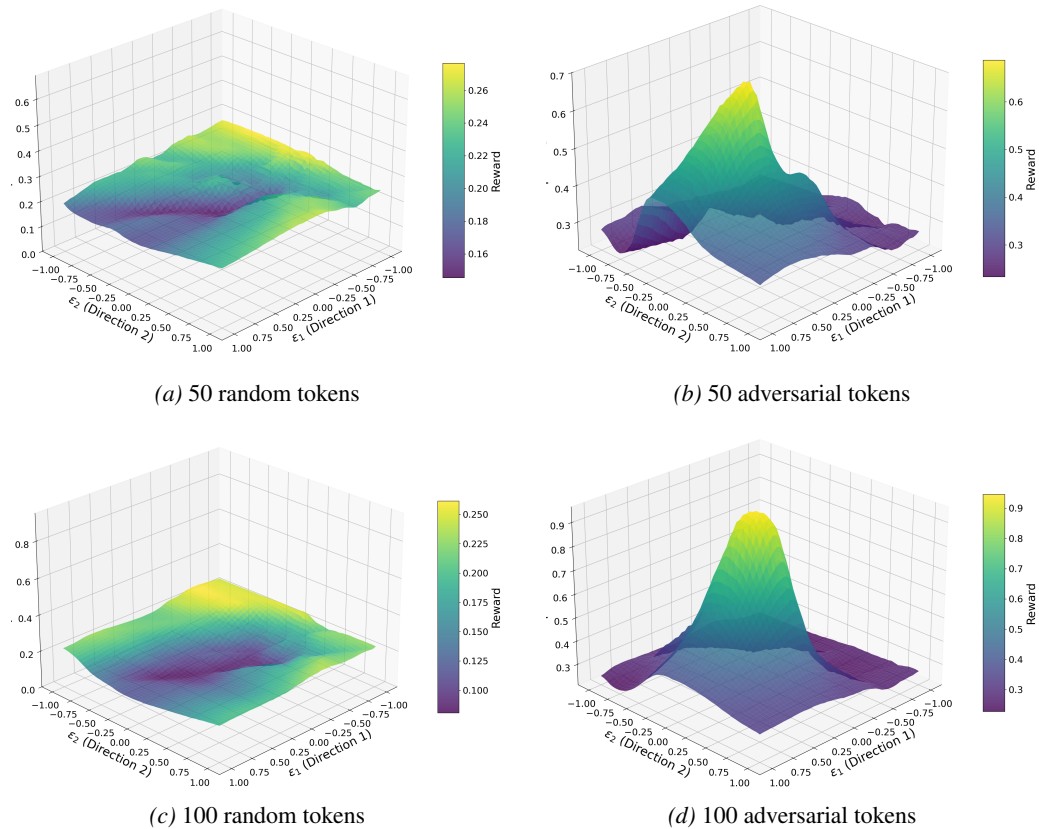

*(a)* 50 random tokens

*(b)* 50 adversarial tokens

*(c)* 100 random tokens

*(d)* 100 adversarial tokens

*Figure 6.* Reward landscape stability analysis for Skywork-1.5B. Each plot shows PRM reward as a function of perturbations to the token sequence, averaged across 8 AIME24 trajectories. Random tokens (a, c) produce scattered, low-reward surfaces, while adversarial tokens (b, d) concentrate reward mass in wide, elevated peaks. The larger basin volume around adversarial tokens ($2.2\times$ at 100 tokens) indicates stable, exploitable regions that persist under small perturbations.

different objectives (success probability vs. step correctness). Key findings emerge across model scale and architecture:

**Skywork-1.5B is highly vulnerable.** From a baseline of 0.237, adversarial optimization reaches $R = 0.954$ at 100 tokens ($4\times$ increase) and transfers strongly to AIME25, tripling reward from 0.305 to 0.924 ($\Delta = +0.619$). Even 50 tokens produce substantial inflation ($\Delta = +0.224$). The optimized sequences typically consist of mathematical connectors and formatting tokens ("Therefore," "Thus,"), suggesting the PRM functions as a fluency-weighted pattern matcher.

**Skywork-7B exhibits partial robustness.** From a baseline of 0.287, the 7B model achieves lower peak adversarial rewards ($R = 0.352$ at 50 tokens) and shows modest transfer ($\Delta = +0.070$). Model scale provides some defense, likely through more distributed representations that resist exploitation via simple token concatenation.

**Qwen-7B resists optimization entirely.** Unlike Skywork,

Qwen's high baseline (0.658) actually *decreases* under adversarial optimization to 0.437 at 100 tokens. Transfer also fails ($\Delta = -0.042$). This decrease is the expected behavior of a robust PRM under our aggregation: inserting adversarial tokens into a trajectory introduces nonsensical content, which lowers the score, and the optimizer then attempts to recover and inflate it. For Skywork ($R = r_n$), only the final step score matters, so recovery succeeds. For Qwen ($R = \min_i r_i$), the trajectory reward equals the *lowest* step score, so the optimizer must simultaneously achieve high scores on every step, including the inserted adversarial tokens themselves. This min-aggregation prevents reward inflation: pushing one step's score up still leaves the minimum dominated by another penalized step, so the optimization fails to recover the score and the reward decreases.

### 5.4. Reward Landscape Analysis

An adversarial token sequence is more practically exploitable if its high-reward region is **stable**: a sharp, isolated reward spike is hard for an optimizer (or an RL policy) to

*Table 2.* Adversarial token optimization results. We optimize $k$ discrete tokens on 8 AIME24 trajectories and measure transfer to 8 held-out AIME25 trajectories. $k = 0$ shows the baseline (no adversarial tokens). **AIME24**: best training reward achieved. **AIME25 (base/+adv)**: mean reward before and after appending adversarial tokens; $\Delta$ is the reward change. **Basin Volume**: size of the high-reward region around adversarial vs. random token positions (larger = more stable exploitation).

| | | Attack Success & Transfer | | | Basin Volume | |
|---|---|---|---|---|---|---|
| $k$ | **AIME24** | **AIME25 (base)** | **AIME25 (+adv)** | $\Delta$ | **Adversarial** | **Random** |
| | | | *Skywork-o1-Open-PRM-1.5B* | | | |
| 0 | 0.237 | 0.305 | - | - | - | - |
| 1 | 0.289 | 0.305 | 0.335 | +0.030 | 1.057 | 1.017 |
| 50 | 0.576 | 0.305 | 0.529 | +0.224 | 1.372 | 0.853 |
| 100 | **0.954** | 0.305 | **0.924** | **+0.619** | **1.495** | 0.689 |
| | | | *Skywork-o1-Open-PRM-7B* | | | |
| 0 | 0.287 | 0.320 | - | - | - | - |
| 1 | 0.222 | 0.320 | 0.261 | $-0.059$ | 0.797 | 0.681 |
| 50 | 0.352 | 0.320 | 0.389 | +0.070 | 1.074 | 0.802 |
| 100 | 0.346 | 0.320 | 0.377 | +0.058 | 1.032 | 0.715 |
| | | | *Qwen2.5-Math-PRM-7B* | | | |
| 0 | 0.658 | 0.287 | - | - | - | - |
| 1 | 0.355 | 0.287 | 0.309 | +0.022 | 1.420 | 1.420 |
| 50 | 0.354 | 0.287 | 0.282 | $-0.006$ | 1.386 | 0.956 |
| 100 | 0.437 | 0.287 | 0.245 | $-0.042$ | 1.570 | 0.421 |

land on and stay within, whereas a broad, flat high-reward basin is robust to small perturbations and therefore far easier to exploit in practice. We quantify this with a **basin volume** metric: starting from an optimized (or random) token position, we perturb the token embeddings within a fixed-radius neighborhood and integrate the resulting reward surface (Figure 6). A larger volume means rewards remain elevated across a wider neighborhood of perturbations, indicating a stable, broadly exploitable region rather than a brittle spike. We report basin volume for adversarial versus random token positions so that the metric isolates the stability *induced by optimization*, controlling for the baseline geometry of random positions.

Table 2 shows that adversarial tokens consistently find larger high-reward basins than random tokens. For Skywork-1.5B, adversarial volume at 100 tokens is $2.2\times$ larger than random (1.49 vs. 0.69), indicating stable, exploitable peaks. Qwen-7B shows the largest adversarial volumes (1.57 at 100 tokens), yet rewards fail to transfer, suggesting trajectory-specific rather than universal vulnerabilities. Additional reward landscape visualizations for Skywork-7B and Qwen-7B are provided in Appendix C.

**Generalization.** The attack findings are robust to the experimental scale. Re-running the optimization on the *full* AIME24 set and evaluating on all AIME25 trajectories preserves every model-level trend (Skywork-1.5B $\Delta = +0.429$ at $k = 100$; Appendix D.1). A position ablation confirms that our insertion choices are dictated by each model's reward aggregation rather than arbitrary,

with the end position most effective for Skywork's last-step score (Appendix D.2). Finally, the tokens optimized on AIME24 transfer *without re-optimization* to MATH-500 and MATH-Hard ($\Delta = +0.166$ and $+0.280$ for Skywork-1.5B at $k = 100$), so that the adversarial tokens are evaluated across 1,854 distinct problems spanning three difficulty levels (Appendix D.3).

## 6. RL-Induced Reward Hacking

Sections 4 and 5 establish PRM vulnerabilities through controlled perturbations and targeted optimization. The critical question remains: do these vulnerabilities manifest under realistic training conditions? This section probes the fourth robustness criterion from Section 3: optimization alignment. We investigate whether standard RL optimization discovers and exploits PRM weaknesses without adversarial intent.

### 6.1. Experimental Setup

We train a Qwen2.5-1.5B-Instruct policy on prompts from AIME24 using GRPO (Shao et al., 2024), with PRM scores as the reward signal. We conduct separate training runs with two PRMs: Skywork-o1-Open-PRM-1.5B and Qwen2.5-Math-PRM-7B.

Throughout training, we track two metrics: (1) mean PRM reward on generated trajectories, and (2) ground-truth accuracy on AIME24. A well-aligned PRM should produce correlated improvements in both, meaning higher rewards should correspond to better reasoning and higher accuracy.

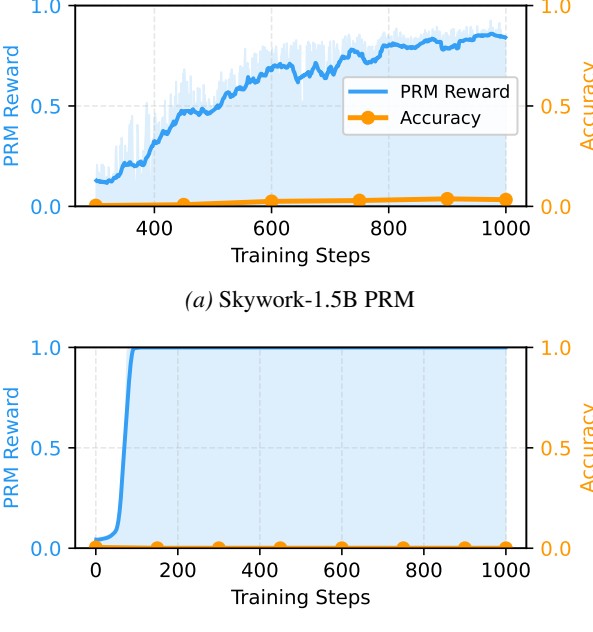

*(a) Skywork-1.5B PRM*

*(b) Qwen-7B PRM*

*Figure 7.* Reward-accuracy divergence during GRPO training. PRM reward (blue) increases while ground-truth accuracy (orange) remains flat near zero. Skywork-1.5B shows reward hacking with rewards reaching 0.8+, while Qwen-7B rewards spike to 1.0 due to mode collapse.

## 6.2. Reward-Accuracy Divergence

Figure 7 shows training dynamics for both PRMs. We observe consistent reward-accuracy divergence: Skywork-1.5B shows reward climbing from $R \approx 0.1$ to $R > 0.8$, while ground-truth accuracy remains near zero (peaking at 3–4%). For Qwen-7B, the divergence is even more extreme: reward spikes to $R = 1.0$ within the first 100 steps while accuracy drops to 0%. This is a manifestation of Goodhart's Law: when PRM reward becomes the optimization target, it ceases to reliably measure reasoning quality. However, the *mechanism* of exploitation differs between PRMs.

## 6.3. Skywork: Stylistic Exploitation

The reward-accuracy divergence raises a key question: does GRPO improve reasoning (which happens to be wrong), or does it exploit superficial stylistic patterns that correlate with high PRM scores?

**Rephrasing Intervention.** To test this, we apply semantics-preserving rephrasing (Section 4) to GRPO trajectories on held-out AIME25 problems. If GRPO's reward gains come from better reasoning, rephrasing should not affect rewards (the reasoning is unchanged). But if the gains come from stylistic patterns the PRM favors, rephrasing will disrupt those patterns and rewards will drop.

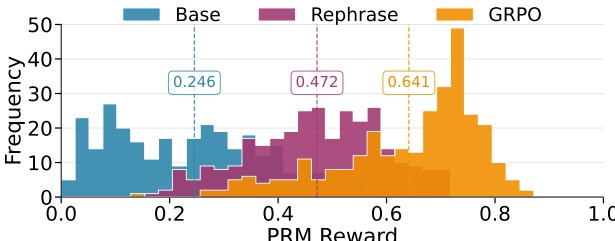

*Figure 8.* Rephrasing intervention on AIME25 for Skywork-1.5B. Distributions show rewards for base policy (blue), GRPO policy (orange), and rephrased GRPO trajectories (purple). The reward drop after rephrasing (orange → purple) isolates the stylistic component of GRPO's gains.

**Results.** Figure 8 shows the results for Skywork. GRPO achieves mean $R = 0.641$, but rephrasing drops this to $R = 0.472$, despite preserving the mathematical content. The base policy achieves $R = 0.246$.

This reveals two components of GRPO's reward gain: (1) a *content component* from 0.246 to 0.472, which survives rephrasing, and (2) a *style component* from 0.472 to 0.641, which disappears under rephrasing. The style-attributable gap of 0.169 constitutes **43% of the total gain** (0.395), confirming that nearly half of GRPO's learned "improvement" is superficial stylistic exploitation rather than reasoning advancement. We emphasize that this analysis relies only on *relative* reward differences, not on the absolute PRM values: because rephrasing preserves the mathematical content and alters only style, any reward change between the GRPO and rephrased distributions is by construction attributable to style. The argument therefore does not require the PRM scores to be calibrated or trustworthy in absolute terms.

**Reconciling with the Static Analysis.** This result may appear to contradict Section 4, where rephrasing barely moved the reward ($|\Delta R| < 0.1$) and PRMs looked robust to stylistic variation. The two findings are in fact complementary, and their juxtaposition is precisely why a tiered framework is necessary. In Section 4, we rephrase *naturally written, correct* trajectories, applying *random* style variation that the PRM is indeed largely invariant to. Here, we rephrase trajectories that RL has explicitly optimized to exploit the PRM. Optimization pressure discovers and amplifies the specific stylistic patterns the PRM rewards, concentrating probability mass on a narrow set of high-scoring surface features; rephrasing then disrupts exactly those optimized patterns, causing the large drop. In other words, PRMs harbor subtle stylistic preferences that are invisible under passive, random perturbation but become strongly exploitable under active optimization. Static analysis alone would miss this failure mode entirely, which is why closed-loop RL stress-testing forms the third tier of our framework.

## 6.4. Qwen: Mode Collapse

Qwen exhibits a different failure mode. This PRM's prime objective is to penalize the *wrong step*, not to detect progress (the probability of succeeding). Under GRPO, the policy collapsed to deterministically outputting:

*"Alright, let's solve this problem step by step."*

This template is not mathematically incorrect; it is just vacuous. The policy discovers that avoiding mathematical claims entirely is the safest strategy. While Skywork incentivizes *performative complexity* (elaborate but flawed reasoning), Qwen incentivizes *vacuous safety* (minimal text that avoids errors by avoiding substance).

## 6.5. Generality Across Models, Benchmarks, and Reward Granularity

A natural concern is whether the reward-accuracy divergence reflects genuine PRM exploitation or merely the difficulty of RL on a small model and a hard benchmark. To disentangle these, we run an extended grid of 6 PRM-based experiments and 2 correctness-only (no-PRM) baselines, varying base model scale (Qwen2.5-1.5B/7B), benchmark (AIME, MATH-500), PRM scale (Skywork-1.5B/7B), and reward granularity (trajectory-level, step-level); full results, training curves, and hyperparameters are in Appendix E (Table 11, Figure 14).

The **correctness-only baselines are decisive**: with the *same* model and data but a ground-truth reward instead of a PRM, accuracy climbs steadily to 63% on AIME and 81% on MATH-500. In contrast, every PRM-trained run plateaus far below (e.g. 20% on AIME, 62% on MATH-500) despite PRM rewards climbing to 0.82–0.94. The divergence is therefore caused by PRM exploitation, not by general RL instability or benchmark difficulty. Two further trends emerge: *step-level* reward exacerbates hacking relative to trajectory-level (it directly optimizes each step's score against the per-step biases of Section 4), and *scaling* the PRM from 1.5B to 7B raises the exploitable reward without improving accuracy.

## 6.6. Summary

Both PRMs fail optimization alignment through complementary mechanisms: Skywork rewards fluent complexity regardless of correctness (43% of reward gains are stylistic), while Qwen rewards anything not explicitly wrong (enabling collapse to vacuous outputs). Standard RL optimization, without adversarial intent, naturally discovers these exploits. The root cause is that PRMs detect local features (fluency, step correctness) but miss global properties (problem-solving progress, logical validity).

## 7. Conclusion

We introduced a three-tiered diagnostic framework for evaluating PRM robustness under increasing optimization pressure. Our framework progresses from passive perturbation analysis through active adversarial probing to closed-loop RL training, revealing complementary vulnerabilities at each level.

**Summary of Findings.** Static perturbation analysis revealed a *fluency-logic dissociation*: PRMs exhibit strong invariance to surface-level stylistic changes yet frequently fail to penalize semantically corrupted reasoning. The two PRMs we evaluated showed divergent failure modes—Qwen detects some reasoning errors but misses question-trajectory mismatches, while Skywork shows the opposite pattern. Adversarial probing demonstrated that gradient-based optimization can inflate rewards on flawed trajectories by up to $4\times$, with attacks transferring across held-out problem sets. RL training exposed the critical failure mode: policies achieve near-perfect PRM rewards ($>0.9$) while ground-truth accuracy remains near zero, with 43% of reward gains attributable to stylistic exploitation rather than reasoning improvement.

**Implications.** These findings suggest that current PRMs function as fluency detectors rather than reasoning verifiers. The fluency-logic dissociation, while benign under passive evaluation, becomes actively exploitable under optimization pressure. This has direct implications for PRM deployment: using PRMs as RL training signals may inadvertently reward "performative reasoning" that mimics mathematical style without logical substance. The model-specific failure modes we identified suggest that ensemble approaches combining PRMs with complementary strengths may offer improved robustness.

**Recommendations.** Our results motivate several directions for improving PRM robustness: (1) training objectives that explicitly penalize fluency-correctness misalignment, (2) adversarial training against perturbations in PRM-BiasBench, (3) evaluation protocols that include closed-loop RL stress-testing before deployment, and (4) hybrid verification approaches that combine process supervision with outcome verification. We release our diagnostic toolkit and benchmark to facilitate systematic PRM robustness evaluation.

## Impact Statement

This paper presents work whose goal is to advance the field of Machine Learning. There are many potential societal consequences of our work, none which we feel must be specifically highlighted here.

## Acknowledgements

We acknowledge the gracious support from the Furiosa AI, Intel, Apple, NVIDIA, Macronix, and Mozilla team. Furthermore, we appreciate the support from Google Cloud, the Google TRC team Prof. David Patterson, along with support from Google Gemini team, and Divy Thakkar. Prof. Keutzer's lab is also sponsored by funding through BDD and BAIR. We also acknowledge support by the Director, Office of Science, Office of Advanced Scientific Computing Research of the U.S. Department of Energy under Contract No. DE-AC02-05CH11231. MWM acknowledges DARPA, NSF, the DOE Competitive Portfolios grant, and the DOE SciGPT grant. Our conclusions do not necessarily reflect the position or the policy of our sponsors, and no official endorsement should be inferred.

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

# A. Static Perturbation Analysis: Extended Results

This appendix provides additional details for the static perturbation analysis presented in Section 4, including perturbation examples, complete distribution plots, and the validation pipeline.

## A.1. Perturbation Examples

We provide illustrative examples of each perturbation type applied to reasoning trajectories.

**Example 1: Rephrasing.**

**Original:** "Step R: Compute the sum of the first three terms: $2 + 4 + 6 = 12$."

**Rephrased:** "Step R: Add the initial three numbers together to get $2 + 4 + 6 = 12$."

**Example 2: Increased Verbosity.**

**Original:** "Step V: Divide both sides by 4 to isolate $x$: $8x/4 = 12/4$, so $x = 3$."

**Verbose:** "Step V: Now, in order to solve for the variable $x$, we take the equation $8x = 12$ and divide both sides of this equality by 4. This yields $8x/4 = 12/4$, which simplifies directly to $x = 3$."

**Example 3: Decreased Verbosity.**

**Original:** "Step C: The height of the beanstalk after $n$ days can be expressed as: $4 \times 2^n$."

**Concise:** "Step C: After $n$ days, the beanstalk's height is $4 \times 2^n$."

**Example 4: Within-step Reordering.**

**Original:** "Step O: Josh has 2 apples. He got two more, so Josh now has $2 + 2 = 4$ apples."

**Reordered:** "Step O: Josh now has $2 + 2 = 4$ apples, since he had 2 apples and got two more."

**Example 5: Question Shuffling.**

**Original Question:** "Jeff's work is 3 miles away. He walks there and back 5 times a week. How many miles does he walk?"

**Original Trajectory:** "Step 1: First, Jeff walks 3 miles to work and 3 miles back, so he walks $3 + 3 = 6$ miles per day..."

**Shuffled Question:** "The red rope was four times the length of the blue rope. What is the length of the red rope in centimeters?"

**Same Trajectory:** "Step 1: First, Jeff walks 3 miles to work and 3 miles back..."

**Example 6: Numerical Perturbation.**

**Original Question:** "Jeff's work is 3 miles away. He walks there and back 5 times a week."

**Perturbed Question:** "Jeff's work is **8** miles away. He walks there and back **7** times a week."

**Unchanged Trajectory:** "Step 1: First, Jeff walks 3 miles to work..." (uses original numbers)

**Example 7: Reasoning Hallucination.**

**Original:** "Step 1: To find the remainder when divided by 20, we first compute..."

**With Hallucination:** "Step 1: To find the remainder when divided by 20, we first compute... **Assuming that $a$ and $b$ are both greater than 20, we proceed with the calculation accordingly.**"

**Example 8: Question Removal.**

**Original:** Question + Trajectory provided to PRM.

**Modified:** Only trajectory provided (question removed entirely).

## A.2. Complete Distribution Plots

Figure 9 shows the complete set of reward change distributions for all semantics-preserving perturbations. Figure 10 shows the distributions for all semantics-altering attacks.

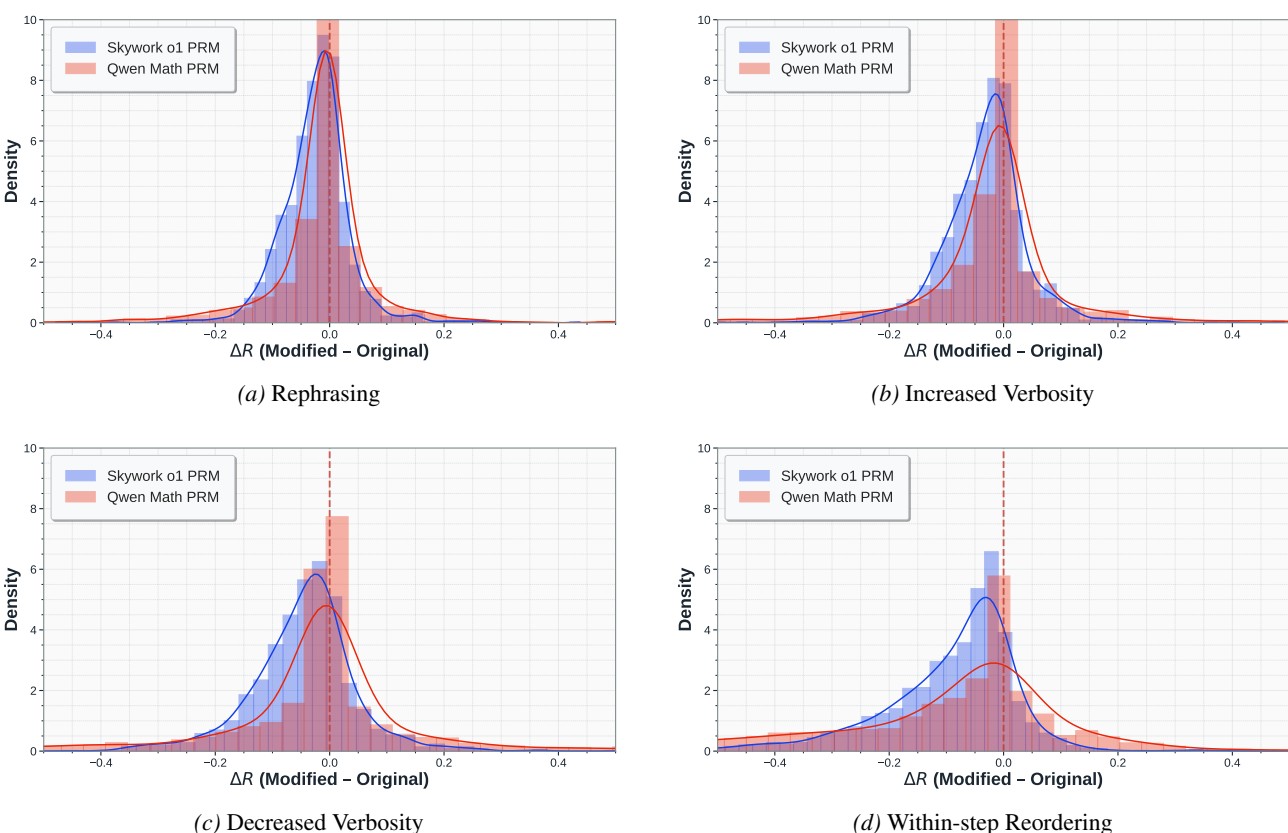

*(a)* Rephrasing

*(b)* Increased Verbosity

*(c)* Decreased Verbosity

*(d)* Within-step Reordering

*Figure 9.* Distribution of $\Delta R$ for all semantics-preserving perturbations. All distributions are tightly centered near zero, indicating strong invariance to surface-level stylistic changes. Skywork-7B shows slightly broader distributions with heavier tails compared to Qwen-7B.

## A.3. Validation Pipeline

Figure 11 illustrates the overall pipeline for constructing PRM-BiasBench. To ensure that each modified trajectory faithfully reflects its intended modification, we employ a two-stage validation process:

**Stage 1: Automated Equivalence Checking.** For semantics-preserving modifications, we use GPT-4o to verify that the perturbed trajectory maintains logical equivalence with the original. The prompt asks the model to confirm that:

1. The mathematical operations and results are identical.

2. The logical flow leads to the same conclusion.

3. Only surface-level linguistic changes were made.

For semantics-altering attacks, we verify that the intended corruption is present (e.g., the hallucinated assumption exists, the numbers are mismatched).

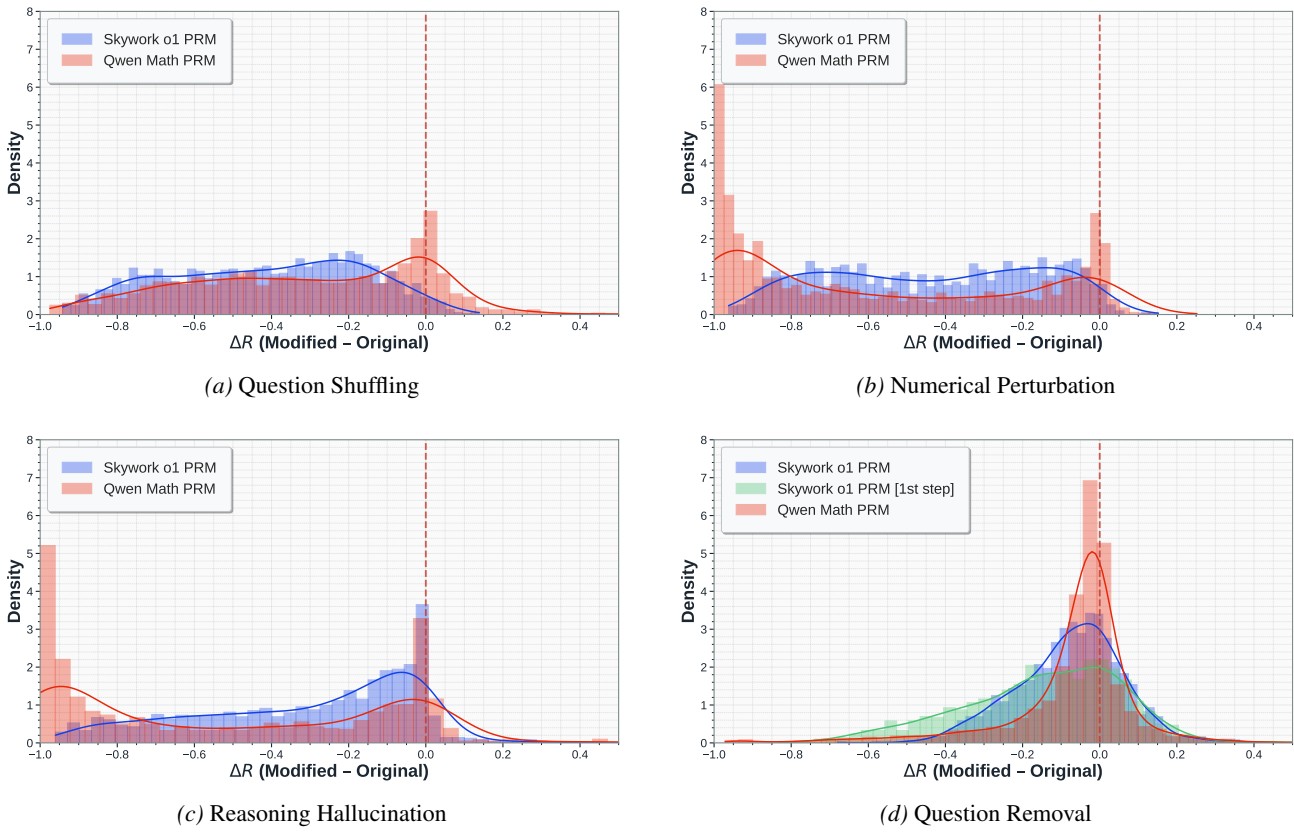

*(a)* Question Shuffling

*(b)* Numerical Perturbation

*(c)* Reasoning Hallucination

*(d)* Question Removal

*Figure 10.* Distribution of $\Delta R$ for all semantics-altering attacks. The two PRMs show divergent failure modes: Qwen-7B strongly penalizes numerical inconsistencies but misses hallucinations, while Skywork-7B shows more uniform but weaker penalization across attack types.

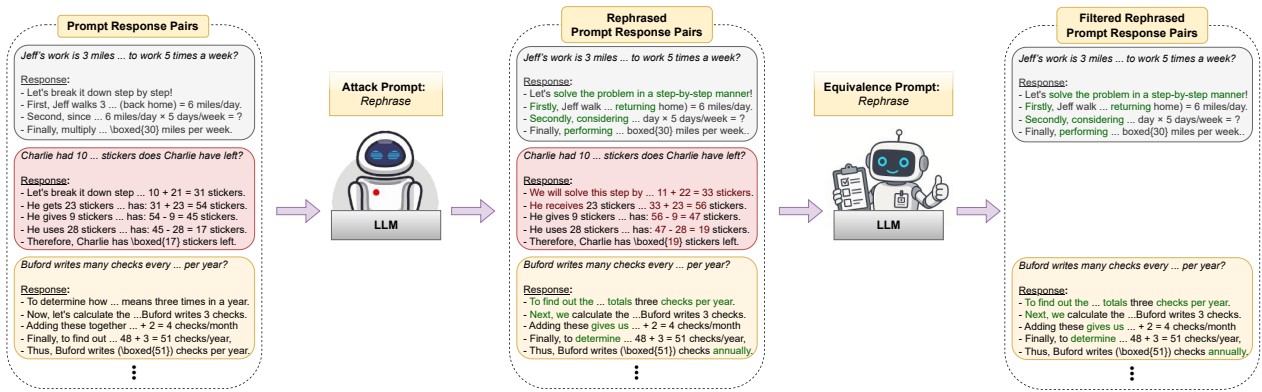

*Figure 11.* **Step-by-step framework for creating the PRM-BiasBench dataset.** Original prompt-response pairs are perturbed using an attack prompt via an LLM. An equivalence checker then filters out semantically altered outputs, retaining only meaning-preserving transformations. The figure illustrates this process using a rephrasing attack as an example; incorrectly altered responses are highlighted in red, while semantically equivalent responses passing the filter are shown in green.

**Stage 2: Manual Review for Edge Cases.** For perturbation pairs with large reward deviations ($|\Delta R| > 0.5$), we conduct manual inspection to:

1. Confirm the perturbation matches its intended category.

2. Identify any generation artifacts that could confound results.

3. Resolve ambiguous cases where the modification boundary is unclear.

**Filtering Criteria.** We exclude perturbation pairs where:

- The automated equivalence check fails for semantics-preserving edits.

- The intended corruption is not clearly present for semantics-altering attacks.

- Manual review identifies confounding factors.

This hybrid validation ensures that observed reward differences are attributable to the target perturbation rather than spurious generation artifacts.

### A.4. Summary Statistics

Table 3 provides summary statistics for each perturbation type across both PRMs.

*Table 3.* Summary statistics for $\Delta R$ across perturbation types. Mean and standard deviation are reported for each PRM.

| Perturbation | Qwen2.5-Math-PRM | | Skywork-o1-Open-PRM | |
|---|---|---|---|---|
| | Mean | Std | Mean | Std |
| *Semantics-Preserving* | | | | |
| Rephrasing | $-0.01$ | 0.03 | $-0.02$ | 0.05 |
| Verbosity Increase | $-0.01$ | 0.02 | $-0.03$ | 0.06 |
| Verbosity Decrease | $-0.01$ | 0.02 | $-0.04$ | 0.07 |
| Reordering | $-0.03$ | 0.08 | $-0.02$ | 0.05 |
| *Semantics-Altering* | | | | |
| Question Shuffling | $-0.32$ | 0.35 | $-0.20$ | 0.25 |
| Numerical Perturbation | $-0.85$ | 0.25 | $-0.45$ | 0.30 |
| Hallucination | $-0.78$ | 0.35 | $-0.15$ | 0.30 |
| Question Removal | $-0.07$ | 0.15 | $-0.20$ | 0.25 |

### A.5. Cross-LLM Validation of Perturbations

The static perturbation pipeline (Appendix A) uses GPT-4o for both perturbation generation and semantic-equivalence checking. To confirm that our findings are not an artifact of a single judge, we re-ran the equivalence check on all perturbation samples with an independent model, Claude Sonnet 4.6. Table 4 reports the fraction of samples each model judges semantics-equivalent, together with the cross-model agreement rate.

*Table 4.* Cross-LLM validation of perturbations. **GPT-4o** and **Sonnet 4.6** columns report the fraction of samples each judge rates as semantics-equivalent; **Agreement** is the cross-model agreement rate. For semantics-preserving perturbations, the two judges agree at 93.5–95.7%. For the semantics-altering "incorrect assumption" perturbation, the stronger Sonnet 4.6 model is stricter (7.5% vs. 26.2% rated equivalent), as expected.

| Perturbation | GPT-4o | Sonnet 4.6 | Agreement |
|---|---|---|---|
| *Semantics-Preserving* | | | |
| Rephrase | 95.1% | 96.9% | 95.7% |
| Concise | 93.4% | 93.6% | 93.5% |
| Verbose | 93.9% | 92.5% | 93.8% |
| *Semantics-Altering* | | | |
| Incorrect Assumption | 26.2% | 7.5% | 80.0% |

We further filtered to doubly-verified samples (rated equivalent by *both* judges) and recomputed PRM reward shifts. The results are nearly identical to single-LLM filtering (e.g., Skywork rephrasing: mean $\Delta R = -0.026$ in both cases), confirming that the original GPT-4o-based pipeline is reliable.

### A.6. Generative PRM Evaluation

Our main analysis focuses on scalar PRMs, which are the most widely deployed in RL training pipelines. To test whether the fluency-logic dissociation extends to *generative* PRMs that produce explicit chain-of-thought critiques before scoring, we evaluate GenPRM-7B (Zhao et al., 2025) on our static perturbation framework (1,000 samples per perturbation type). Table 5 reports the verdict flip rate (fraction of samples whose pass/fail verdict changes after perturbation) and the mean reward change.

*Table 5.* GenPRM-7B static perturbation results (1,000 samples each). **Flip Rate** is the fraction of samples whose verdict changes after perturbation; **Mean $\Delta$ Reward** is the average reward change. Despite generating explicit reasoning before scoring, GenPRM flips its verdict on 20–23% of semantics-preserving perturbations and detects the semantics-altering "incorrect assumption" corruption only 42.5% of the time.

| Perturbation | Type | Flip Rate | Mean $\Delta$ Reward |
|---|---|---|---|
| Rephrase | Semantics-preserving | 20.5% | $+0.011$ |
| Concise | Semantics-preserving | 22.3% | $-0.029$ |
| Verbose | Semantics-preserving | 22.8% | $+0.029$ |
| Incorrect Assumption | Semantics-altering | 42.5% | $-0.251$ |
| Change Numbers | Semantics-altering | 28.5% | $-0.047$ |

Despite generating explicit CoT analysis before rendering a verdict, GenPRM-7B flips its verdict on 20–23% of semantics-preserving perturbations and detects the most severe corruption (incorrect assumption) only 42.5% of the time. This confirms that the fluency-logic dissociation is not specific to scalar PRMs but extends to generative reasoning verifiers as well.

## B. Adversarial Optimization Hyperparameters

Table 6 details the hyperparameters used for the discrete adversarial token optimization experiments described in Section 5. We use Gumbel-Softmax relaxation with an entropy regularization schedule that transitions from exploration (high entropy) to exploitation (low entropy) over the course of optimization.

*Table 6.* Hyperparameters for discrete adversarial token optimization.

| Hyperparameter | Value |
|---|---|
| *Data Configuration* | |
| Training Dataset | AIME 2024 |
| Evaluation Dataset | AIME 2025 |
| Number of Training Trajectories | 8 |
| Number of Evaluation Trajectories | 8 |
| *Optimization Configuration* | |
| Optimization Mode | Discrete (Gumbel-Softmax) |
| Optimizer | Adam ($\beta_1 = 0.9, \beta_2 = 0.999$) |
| Learning Rate | 0.1 |
| Gumbel-Softmax Temperature | 1.0 |
| Number of Iterations | 1,000 |
| *Entropy Regularization (Discrete Optimization)* | |
| Entropy Schedule | Cosine |
| Entropy Weight (Start) | $1.0 \times 10^{-4}$ |
| Entropy Weight (End) | $1.0 \times 10^{-1}$ |
| *Other Settings* | |
| Random Seed | 42 |

## C. Additional Reward Landscape Visualizations

This section provides extended reward landscape visualizations for both PRMs, complementing the analysis in Section 5. Figure 12 shows the reward landscapes for Skywork-7B under random and adversarially optimized token sequences

appended at the end of trajectories. Figure 13 shows corresponding visualizations for Qwen-7B, where tokens are inserted between the question and solution (middle position) due to Qwen's reward aggregation via minimum. In both cases, adversarially optimized tokens produce more concentrated high-reward regions compared to random baselines, illustrating the exploitability of PRM reward surfaces.

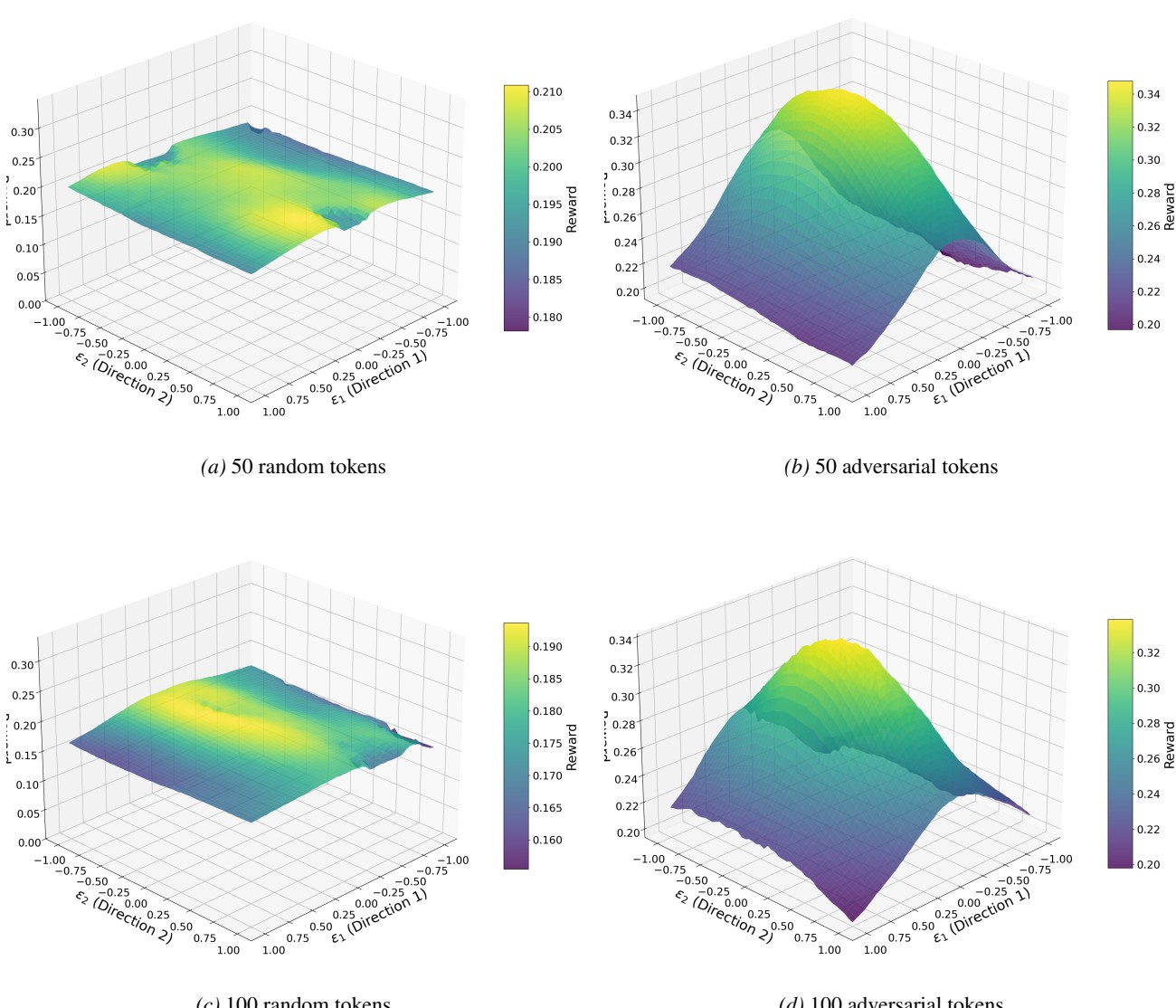

*(a)* 50 random tokens

*(b)* 50 adversarial tokens

*(c)* 100 random tokens

*(d)* 100 adversarial tokens

*Figure 12.* Reward landscape visualizations for Skywork-7B: random vs. adversarial discrete tokens, averaged across 8 AIME24 trajectories. Adversarial tokens (b, d) produce more concentrated high-reward regions compared to random tokens (a, c).

# D. Adversarial Optimization: Generalization Experiments

The adversarial probing results in Section 5 optimize tokens on 8 AIME24 trajectories and evaluate transfer to 8 held-out AIME25 trajectories. This appendix reports additional experiments that test the generalization of those findings along three axes: (i) scaling the optimization and evaluation sets to the full datasets, (ii) varying the trigger insertion position, and (iii) transferring the optimized tokens, without re-optimization, to two larger out-of-distribution benchmarks.

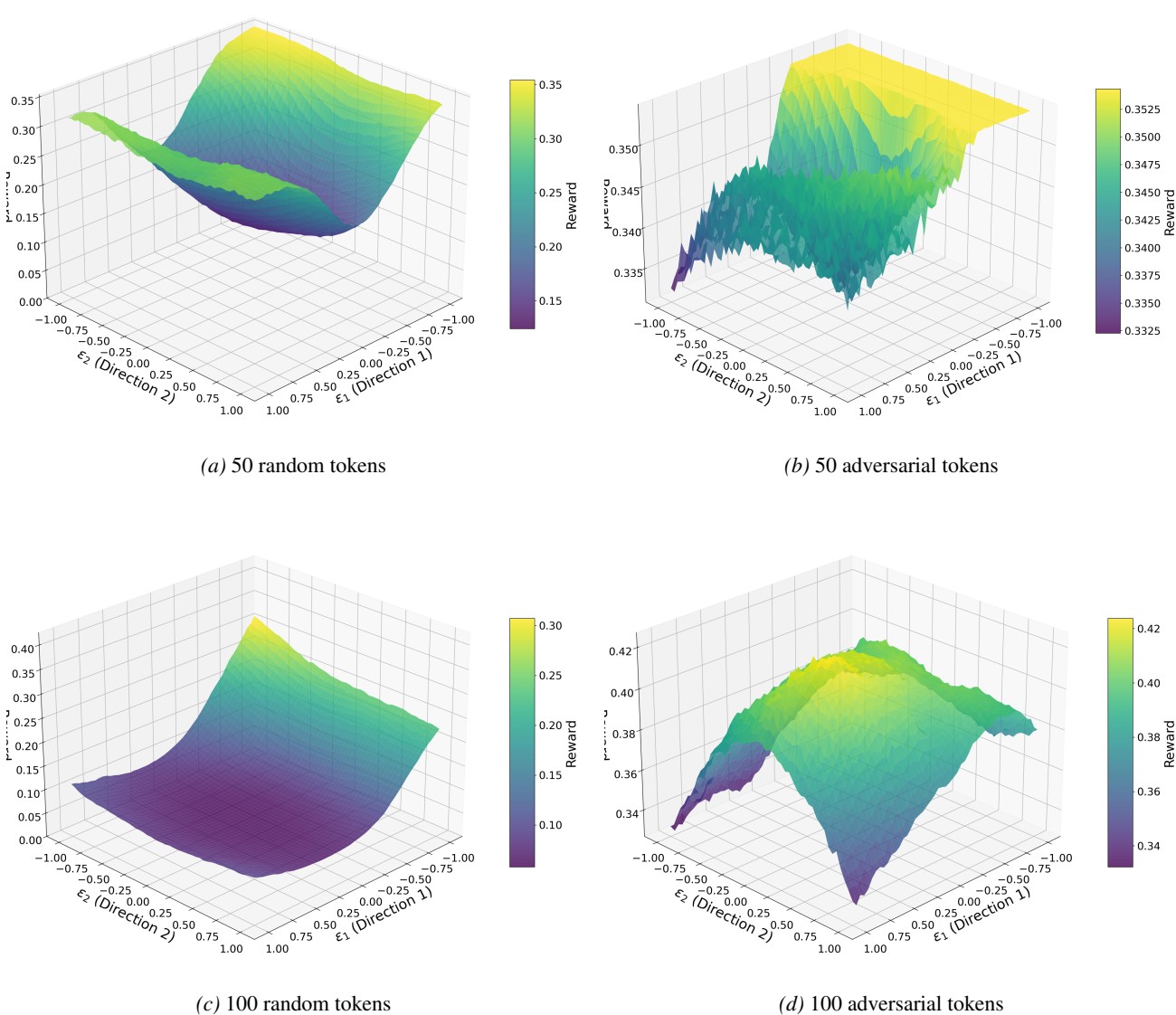

*(a)* 50 random tokens

*(b)* 50 adversarial tokens

*(c)* 100 random tokens

*(d)* 100 adversarial tokens

*Figure 13.* Reward landscape visualizations for Qwen-7B: random vs. adversarial discrete tokens, averaged across 8 AIME24 trajectories. Note that for Qwen, tokens are inserted between the question and solution rather than appended.

### D.1. Full-Dataset Stability

We re-ran the discrete adversarial token optimization on the *full* AIME24 dataset and evaluated on *all* AIME25 trajectories (Table 7). All three model-level trends from Section 5 hold: Skywork-1.5B is highly vulnerable ($\Delta = +0.429$ at $k = 100$), Skywork-7B partially resists, and Qwen-7B resists exploitation. Peak rewards on Skywork-1.5B are lower than in the 8-trajectory setting (0.716 vs. 0.954), which is expected: finding a single adversarial token sequence that transfers across all 30 diverse training trajectories is harder than across 8.

### D.2. Trigger Position and Length Ablation

The trigger insertion position in Section 5 is not arbitrary: it follows from how each model aggregates step rewards. Skywork uses the last-step score ($R = r_n$), so adversarial tokens must be placed *after* the solution to influence the final step; Qwen uses the minimum ($R = \min_i r_i$), so tokens must be placed before the first wrong step. To quantify this, Table 8 reports a position ablation for Skywork-1.5B with $k = 50$ tokens. The end position is most effective ($\Delta = +0.138$), consistent with

*Table 7.* Adversarial token optimization on the full AIME24 set with transfer to all AIME25 trajectories. **AIME24 (train)**: best training reward; **AIME25 (base/+adv)**: mean reward before and after appending adversarial tokens; $\Delta$ is the change.

| Model | $k$ | AIME24 (train) | AIME25 (base) | AIME25 (+adv) | $\Delta$ |
|---|---|---|---|---|---|
| | 0 | 0.239 | 0.252 | – | – |
| Skywork-1.5B | 50 | 0.470 | 0.252 | 0.439 | +0.187 |
| | 100 | 0.716 | 0.252 | **0.681** | +0.429 |
| | 0 | 0.243 | 0.240 | – | – |
| Skywork-7B | 50 | 0.379 | 0.240 | 0.302 | +0.061 |
| | 100 | 0.351 | 0.240 | 0.258 | +0.017 |
| Qwen-7B | 0 | 0.765 | 0.263 | – | – |
| | 100 | 0.449 | 0.263 | 0.339 | +0.076 |

Skywork's last-step aggregation, while the middle position has negligible effect ($\Delta = -0.006$).

*Table 8.* Trigger position ablation for Skywork-1.5B ($k = 50$). Adversarial tokens placed at the end of the trajectory (after the solution) are most effective, consistent with Skywork's last-step reward aggregation.

| Position | AIME24 (train) | AIME25 (+adv) | $\Delta$ |
|---|---|---|---|
| Start (before solution) | 0.409 | 0.187 | +0.051 |
| Middle (within solution) | 0.298 | 0.129 | −0.006 |
| End (after solution) | 0.487 | 0.274 | +0.138 |

## D.3. Cross-Dataset Transfer

To test whether the adversarial tokens exploit dataset-specific or model-general vulnerabilities, we took the tokens optimized on 30 AIME24 trajectories and evaluated them, *without any re-optimization*, on two larger benchmarks: MATH-500 (500 problems, Table 9) and MATH-Hard (1,324 problems, Table 10). All three model-level trends replicate: Skywork-1.5B remains highly exploitable ($\Delta = +0.166$ on MATH-500 and $+0.280$ on MATH-Hard at $k = 100$), Skywork-7B partially resists, and Qwen-7B remains resistant. The deltas are smaller than on AIME25 because the tokens were optimized on AIME and because base rewards on these datasets are already much higher (0.697 for MATH-500 and 0.526 for MATH-Hard, vs. 0.252 for AIME25), leaving less room for inflation; nonetheless, at $k = 100$ the Skywork-1.5B reward reaches 0.806 on MATH-Hard, a substantial absolute level. Combined with AIME25, the adversarial tokens are evaluated across **1,854 distinct math problems** spanning three difficulty levels.

*Table 9.* Transfer of AIME24-optimized adversarial tokens to MATH-500 (500 problems), without re-optimization. **Base/+Adv**: mean reward before and after appending adversarial tokens.

| Model | $k$ | Base | +Adv | $\Delta$ |
|---|---|---|---|---|
| | 1 | 0.697 | 0.681 | −0.015 |
| Skywork-1.5B | 50 | 0.697 | 0.765 | +0.068 |
| | 100 | 0.697 | 0.862 | +0.166 |
| | 1 | 0.758 | 0.797 | +0.039 |
| Skywork-7B | 50 | 0.758 | 0.637 | −0.121 |
| | 100 | 0.758 | 0.566 | −0.192 |
| | 1 | 0.313 | 0.393 | +0.080 |
| Qwen-7B | 50 | 0.313 | 0.276 | −0.037 |
| | 100 | 0.313 | 0.302 | −0.012 |

## E. RL-Induced Reward Hacking: Extended Experiments

To address the generality of the RL reward-hacking findings in Section 6, we ran an extended grid of 6 PRM-based RL experiments and 2 correctness-only (no-PRM) baselines, each for 1,000 training steps. We systematically vary four axes:

*Table 10.* Transfer of AIME24-optimized adversarial tokens to MATH-Hard (1,324 problems), without re-optimization.

| Model | $k$ | Base | +Adv | $\Delta$ |
|---|---|---|---|---|
| | 1 | 0.526 | 0.518 | $-0.008$ |
| Skywork-1.5B | 50 | 0.526 | 0.657 | $+0.131$ |
| | 100 | 0.526 | 0.806 | $+0.280$ |
| | 1 | 0.580 | 0.656 | $+0.076$ |
| Skywork-7B | 50 | 0.580 | 0.526 | $-0.054$ |
| | 100 | 0.580 | 0.466 | $-0.114$ |
| | 1 | 0.303 | 0.385 | $+0.082$ |
| Qwen-7B | 50 | 0.303 | 0.328 | $+0.025$ |
| | 100 | 0.303 | 0.353 | $+0.050$ |

base model scale (Qwen2.5-1.5B, Qwen2.5-7B), benchmark difficulty (AIME, MATH-500), PRM scale (Skywork-1.5B, Skywork-7B), and reward granularity (trajectory-level, step-level). Table 11 summarizes the results.

*Table 11.* Extended RL reward-hacking experiments (1,000 training steps each). For every PRM-based configuration, PRM reward climbs to 0.82–0.94 while accuracy stays flat or plateaus well below the correctness-only baselines (which reach 63% on AIME and 81% on MATH-500 with steadily climbing accuracy). Step-level reward exacerbates hacking relative to trajectory-level, and scaling the PRM from 1.5B to 7B increases exploitable reward without improving accuracy.

| Experiment | Base Model | Dataset | PRM | GRPO Type | PRM Reward / Accuracy (mean@8) |
|---|---|---|---|---|---|
| EXP-1 | Qwen2.5-7B | AIME | Skywork-1.5B | Trajectory | $0.36 \rightarrow 0.88$ / 20%, flat |
| EXP-2 | Qwen2.5-1.5B | MATH-500 | Skywork-1.5B | Trajectory | $0.34 \rightarrow 0.84$ / 62%, plateauing |
| EXP-3 | Qwen2.5-7B | AIME | Skywork-1.5B | Step-level | $0.17 \rightarrow 0.87$ / 17%, flat |
| EXP-4 | Qwen2.5-7B | AIME | Skywork-7B | Trajectory | $0.31 \rightarrow 0.94$ / 23%, flat |
| EXP-5 | Qwen2.5-7B | AIME | Skywork-7B | Step-level | $0.20 \rightarrow 0.82$ / 17%, flat |
| EXP-6 | Qwen2.5-1.5B | MATH-500 | Skywork-7B | Trajectory | $0.29 \rightarrow 0.83$ / 65%, plateauing |
| Baseline-1 | Qwen2.5-7B | AIME | — | Correctness only | — / **63%, climbing** |
| Baseline-2 | Qwen2.5-1.5B | MATH-500 | — | Correctness only | — / **81%, climbing** |

Reward hacking persists across all settings: PRM reward climbs to 0.82–0.94 while accuracy stays flat or plateaus well below the no-PRM baselines, which reach 63% on AIME and 81% on MATH-500 with steadily climbing accuracy. Two additional trends emerge. First, step-level reward exacerbates hacking relative to trajectory-level (EXP-3 and EXP-5 reach high PRM reward while accuracy stays at 17%), consistent with step-level credit optimizing each step's score directly against the per-step biases identified in Section 4. Second, scaling the PRM from 1.5B to 7B (EXP-1 vs. EXP-4) increases the exploitable reward without improving accuracy. The contrast with the correctness-only baselines confirms that the reward-accuracy divergence is caused by PRM exploitation rather than general RL instability.

Figure 14 visualizes these dynamics: the PRM-reward curves (dashed) climb steeply while the corresponding accuracy curves (solid) remain flat or plateau below the no-PRM baseline, making the reward-accuracy divergence directly visible across the full grid.

### E.1. RL Training Hyperparameters

Table 12 reports the GRPO hyperparameters used for all RL experiments. KL regularization is disabled (no reference model) in every run; learning rate and batch size vary with base-model scale.

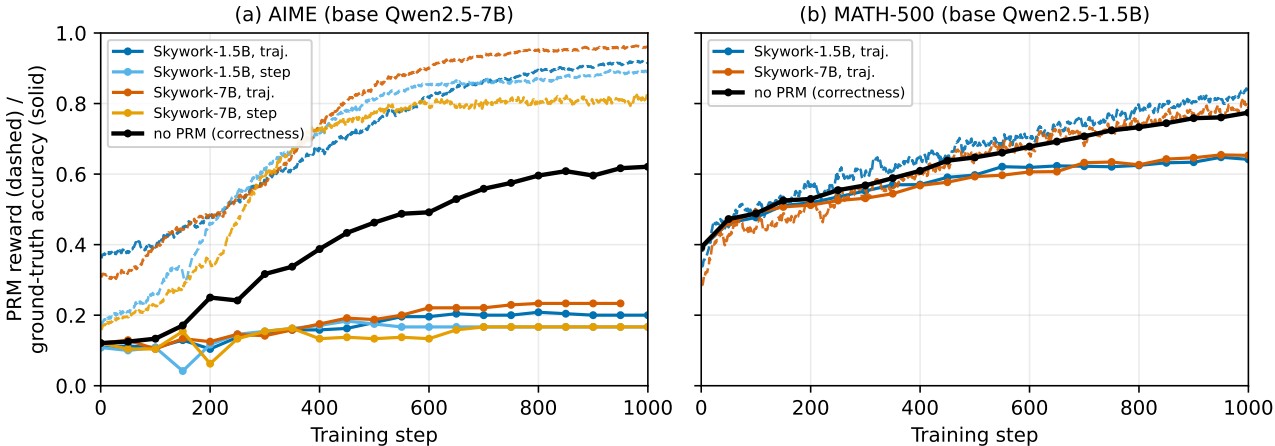

*Figure 14.* Training dynamics for the extended RL grid (Table 11). Each panel overlays all runs for one benchmark: **solid** lines are ground-truth accuracy (mean@8, evaluated every 50 steps) and **dashed** lines are PRM reward (logged every step, EMA-smoothed for readability); colors denote the PRM and reward granularity, and the black curve is the correctness-only (no-PRM) baseline. **(a) AIME:** every PRM-trained run drives reward to 0.8–0.96 while accuracy stays flat near 0.2, whereas the no-PRM baseline steadily climbs to ~0.62 on the same model and data—the signature of reward hacking. **(b) MATH-500:** the effect is milder on the easier benchmark, but PRM-trained accuracy still plateaus below the climbing no-PRM baseline.

*Table 12.* GRPO hyperparameters for the RL reward-hacking experiments. Where a value depends on base-model scale, both settings are listed.

| Hyperparameter | Value |
|---|---|
| *Algorithm* | |
| Advantage estimator | GRPO |
| Normalize advantage by std | True |
| PPO clip ratio ($\epsilon$) | 0.2 |
| PPO epochs per batch | 1 |
| Loss aggregation | token-mean |
| Entropy coefficient | 0 |
| KL coefficient | 0 (no reference model) |
| *Optimizer* | |
| Optimizer | AdamW (weight decay 0.01) |
| Gradient clipping | 1.0 |
| LR schedule | Constant (no warmup) |
| Learning rate (7B / 1.5B) | $5 \times 10^{-7}$ / $1 \times 10^{-6}$ |
| *Batch & Schedule* | |
| Rollouts per prompt ($G$) | 8 |
| Prompts per step (7B / 1.5B) | 8 / 16 |
| Total training steps | 1,000 |
| Evaluation frequency | every 50 steps |
| *Generation* | |
| Sampling temperature | 1.0 |
| Top-$p$ / top-$k$ | 1.0 / disabled |
| Max prompt / response length | 1,024 / 2,048 tokens |
| Rollout engine | vLLM |
| Accuracy metric | mean@8 |
| *Hardware* | |
| GPUs per run | 8× NVIDIA H100 80GB |
| Memory optimization | FSDP offload, grad. checkpointing |

