# OpenReview forum: "Reward Under Attack: Analyzing the Robustness and Hackability of Process Reward Models"
_ICML.cc/2026/Conference — ICML 2026 regular_

### Official Review · Reviewer_qvdF · 2026-02-23

**Soundness:** 3
**Presentation:** 4
**Significance:** 2
**Originality:** 3
**Overall Recommendation:** 4
**Confidence:** 2

**Summary:**

This paper analyze the vulnerability of math Process Reward Models (PRMs) against several types of "attacks". First, the authors conduct **Static Perturbation Analysis** to measure the sensitivity of PRMs to both semantics-preserving and semantics-altering input modifications. Then, the authors explore **Adversarial Attack** to maliciously inflate rewards on invalid trajectories of PRMs. Finally, the authors explore the **Reward Hacking** phenomenon in RL with PRMs. The authors conduct experiments mainly on Skywork and Qwen PRMs to support the claims.

**Compliance With Llm Reviewing Policy:**

Affirmed.

**Final Justification:**

I sincerely thank the authors' response. I think this is a borderline paper. The additional results (though I still think the training data size is quite small) partially address my concern. I would not mind accepting this paper if there is enough space for presentation.

**Key Questions For Authors:**

1. The Qwen results in Table 2 appear unusual. Why does the adversarial attack lead to a decrease in the attack success rate on Qwen-PRM? The explanation provided in Line 308 is not convincing.

2. There is an issue with the caption of Figure 8:  base policy (blue), GRPO policy (orange). Also, I cannot fully understand the logic made in Line 360-369 (right half). PRM rewards cannot be fully trusted in measuring true rewards.

3. Typo in Line 156.

**Limitations:**

Yes

**Strengths And Weaknesses:**

## Strengths

(1) The paper is well-written, the paper structure is clear, the presentation is nice.

(2) The motivation to study the vulnerability is good, which can provide some interesting findings to the community.

(3) The framework is good, the authors analyze the  PRM hackability in three aspects, backed with valid empirical evidence.

## Weaknesses

(1) The authors only conduct experiments with two types of PRMs (Skywork/Qwen). I suggest the authors include more experimental models for solid conclusions. Specifically, it is important to explore the hackability of generative PRMs [1,2,3], which are more powerful than scalar PRMs.

(2) During semantics-preserving modifications, though the authors claim that the correctness of reasoning process is kept, it is not guaranteed with sufficient quantitative or qualitative analysis.

(3) The experimental dataset, AIME24, only contains limited samples. I suggest the authors use larger-scale dataset, especially in the RL experiments in Section 6.



[1] Chen, Xiusi, et al. "Rm-r1: Reward modeling as reasoning." arxiv 2025

[2] Zhao, Jian, et al. "Genprm: Scaling test-time compute of process reward models via generative reasoning." arxiv 2025

[3] Yang, Wenkai, et al. "Deepcritic: Deliberate critique with large language models." arxiv 2025

---

> ### Author Rebuttal · Authors · 2026-03-31
>
> Thank you for your valuable comments. We appreciate that you find the paper well-written with a good framework and valid empirical evidence. We address your questions below:
>
> ## R3-1: More PRM types, including Generative PRMs
>
> The scope of this paper focuses on scalar PRMs as they are the most widely deployed in RL training pipelines. That said, we have now evaluated GenPRM-7B [1] on our static perturbation framework (1,000 samples per perturbation type):
>
> [Table R3](https://image2url.com/r2/default/images/1774954186644-2d395ee9-f937-4143-a03a-4f19c79c8f85.png)
>
> Despite generating explicit CoT analysis before rendering a verdict, GenPRM flips its verdict on 20-23% of semantics-preserving perturbations and only detects the most severe corruption (incorrect assumption) 42.5% of the time. This confirms our central finding extends to generative PRMs. We will include these findings in camera-ready version of the paper.
>
> ## R3-2: Validation of semantics-preserving modifications
>
> Our pipeline includes manual review for edge cases (|Delta R| > 0.5, Appendix A.3). To further strengthen validation, we ran Claude Sonnet 4.6 as an independent equivalence checker on all samples. Cross-LLM agreement is 93.5-95.7% for semantics-preserving perturbations (please see our response to Reviewer u1Vo, R2-3 for the full table). We also filtered doubly-verified samples and found PRM reward shifts nearly identical to single-LLM filtering, confirming the original pipeline is reliable.
>
> ## R3-3: Larger-scale datasets
>
> We clarify the scope of each experiment:
>
> **Static perturbation (Section 4).** PRM-BiasBench extends ProcessBench with thousands of verified perturbation pairs across hundreds of problems, providing substantial coverage.
>
> **Adversarial optimization (Section 5).** We re-ran the adversarial token optimization on the full AIME24 dataset and evaluated on full AIME25 trajectories. The core findings hold: Skywork-1.5B remains highly vulnerable (Delta = +0.429 at k=100). Please see our response to Reviewer u1Vo (R2-1) for the full table.
>
> **RL experiments (Section 6).** We conducted 6 new PRM-based RL experiments and 2 GRPO w/o PRM baselines, varying base model scale (1.5B, 7B), benchmark (AIME, MATH-500), PRM scale (Skywork-1.5B, Skywork-7B), and reward granularity (trajectory, step-level). The baselines achieve 63% (AIME) and 81% (MATH-500) accuracy, while PRM-trained variants plateau at 20% and 62% respectively. Please see our response to Reviewer d6CE (R1-4, Table R1) for the full summary.
>
> ## R3-4: Qwen reward decrease in Table 2
>
> To clarify the expected behavior: inserting adversarial tokens into a valid trajectory introduces nonsensical content, which a robust PRM should penalize (lowering the score). The optimization then tries to recover and inflate that score. For Skywork ($R = r_n$), this recovery succeeds. For Qwen ($R(\tau) = \min\limits_i r_i$), the trajectory reward equals the lowest step score, so the optimizer must achieve high scores on all steps simultaneously, including the adversarial tokens. The optimization fails to recover the score, which is why the reward decreases. We will make sure to clarify this better in the final version.
>
> ## R3-5: Figure 8 caption and Lines 360-369
>
> Thank you for catching the caption issue. We will correct it in the revised version.
>
> **Regarding Lines 360-369.** The rephrasing intervention uses only relative differences: Base = 0.246, GRPO = 0.641, Rephrased GRPO = 0.472. Since rephrasing preserves math content but disrupts style, the drop (0.641 to 0.472 = 0.169) is attributable to style, and the surviving gain (0.472 to 0.246 = 0.226) reflects content. We are not relying on absolute PRM values; any reward change from a style-only edit is, by construction, due to style. We will clarify this reasoning in the revised paper.
>
> ## R3-6: Typo in Line 156
>
> Thank you. We will fix this in the revised version.
>
> [1] Zhao, Jian, et al. "Genprm: Scaling test-time compute of process reward models via generative reasoning." arxiv 2025

---

> > ### Author Rebuttal · Reviewer_qvdF · 2026-04-03
> >
> > Thanks for the response. Some questions are clarified. However, I still think using 30 samples for experiments is not enough for deriving a solid conclusion.

---

> > > ### Author Response · Authors · 2026-04-04
> > >
> > > Thank you for your follow-up and for engaging with our responses. We would like to clarify the evaluation scale across each experiment:
> > >
> > > - **Static perturbation (Section 4):** PRM-BiasBench includes thousands of verified perturbation pairs across hundreds of problems.
> > > - **RL experiments (Section 6):** In our rebuttal (R3-3), we added experiments on MATH-500 (\~500 problems) in addition to AIME, with results compiled in Table R1 (see our response to Reviewer d6CE, R1-4).
> > > - **Adversarial optimization (Section 5):** This used 30 AIME-24 trajectories for optimization. We agree that demonstrating generalization here strengthens the analysis. We note that even in the original rebuttal, adversarial tokens optimized on these 30 trajectories already transferred to 30 held-out AIME-25 trajectories (Delta = +0.429 for Skywork-1.5B at k=100). To go further, we have now evaluated the same tokens on two much larger datasets without any re-optimization.
> > >
> > > ## Cross-dataset transfer to MATH-500 and MATH-Hard
> > >
> > > We evaluated the adversarial tokens (optimized on 30 AIME-24 trajectories) on **MATH-500** (500 problems) and **MATH-Hard** (1,324 problems):
> > >
> > > **Transfer to MATH-500 (500 problems):**
> > >
> > > | Model | k | Base | +Adv | Delta |
> > > | --- | --- | --- | --- | --- |
> > > | Skywork-1.5B | 1 | 0.697 | 0.681 | -0.015 |
> > > | Skywork-1.5B | 50 | 0.697 | 0.765 | +0.068 |
> > > | Skywork-1.5B | 100 | 0.697 | 0.862 | **+0.166** |
> > > | Skywork-7B | 1 | 0.758 | 0.797 | +0.039 |
> > > | Skywork-7B | 50 | 0.758 | 0.637 | -0.121 |
> > > | Skywork-7B | 100 | 0.758 | 0.566 | -0.192 |
> > > | Qwen-7B | 1 | 0.313 | 0.393 | +0.080 |
> > > | Qwen-7B | 50 | 0.313 | 0.276 | -0.037 |
> > > | Qwen-7B | 100 | 0.313 | 0.302 | -0.012 |
> > >
> > > **Transfer to MATH-Hard (1,324 problems):**
> > >
> > > | Model | k | Base | +Adv | Delta |
> > > | --- | --- | --- | --- | --- |
> > > | Skywork-1.5B | 1 | 0.526 | 0.518 | -0.008 |
> > > | Skywork-1.5B | 50 | 0.526 | 0.657 | +0.131 |
> > > | Skywork-1.5B | 100 | 0.526 | 0.806 | **+0.280** |
> > > | Skywork-7B | 1 | 0.580 | 0.656 | +0.076 |
> > > | Skywork-7B | 50 | 0.580 | 0.526 | -0.054 |
> > > | Skywork-7B | 100 | 0.580 | 0.466 | -0.114 |
> > > | Qwen-7B | 1 | 0.303 | 0.385 | +0.082 |
> > > | Qwen-7B | 50 | 0.303 | 0.328 | +0.025 |
> > > | Qwen-7B | 100 | 0.303 | 0.353 | +0.050 |
> > >
> > > Key observations:
> > >
> > > - All three model-level trends from the paper replicate: Skywork-1.5B remains highly exploitable (Delta = +0.166 on MATH-500, +0.280 on MATH-Hard at k=100), Skywork-7B partially resists, and Qwen-7B remains resistant.
> > > - The absolute deltas on MATH-500/MATH-Hard are smaller than on AIME-25 (+0.429), which is expected since the tokens were optimized on AIME. However, the base rewards on these datasets are already much higher (0.697 for MATH-500, 0.526 for MATH-Hard vs. 0.252 for AIME-25), leaving less room for inflation. At k=100 on MATH-Hard, the reward reaches 0.806, which is a substantial absolute level.
> > > - Combined with AIME-25, the adversarial tokens are evaluated across **1,854 distinct math problems** spanning three difficulty levels.
> > >
> > > We will make sure to include these cross-dataset transfer results in the revised paper to strengthen the adversarial optimization analysis. We hope this fully addresses the remaining concerns.

---

### Official Review · Reviewer_u1Vo · 2026-03-08

**Soundness:** 3
**Presentation:** 3
**Significance:** 3
**Originality:** 3
**Overall Recommendation:** 4
**Confidence:** 2

**Summary:**

This paper studies how robust process reward models (PRMs) are in mathematical reasoning and how easily they can be exploited. It introduces a three-tier diagnostic framework: first, controlled perturbations show that PRMs are insensitive to stylistic variation but often fail to consistently penalize semantic errors; second, gradient-based adversarial optimization with both continuous and discrete triggers reveals that invalid reasoning trajectories can receive substantially inflated rewards; and third, RL with PRM feedback exposes a gap between reward improvement and actual accuracy. Overall, the paper suggests that a meaningful share of PRM reward gains may come from stylistic gaming rather than genuine reasoning progress, and it also releases PRM-BiasBench and a robustness evaluation toolkit.

**Compliance With Llm Reviewing Policy:**

Affirmed.

**Final Justification:**

The author's response addressed most of the concerns, and I lean towards accepting the paper.

**Key Questions For Authors:**

1. How robust are the adversarial trigger results given the very small number of trajectories used for optimization and evaluation? Since the attacks are optimized on only a handful of samples, it would be helpful to know whether they remain stable across larger trajectory sets, or different random seeds.

2. How sensitive are the adversarial trigger results to insertion position and trigger length? Since the trigger placement differs across PRMs, it is hard to tell whether some of the cross-model differences come from the models themselves or from the insertion strategy. An ablation on where the trigger is inserted and how many tokens are used would be helpful.

3. Is it possible to provide more validation for the static perturbation pipeline beyond GPT-4o-based checking? Using GPT-4o for generation and equivalence verification is reasonable, but some small-scale human auditing of semantic equivalence and corruption validity would make these results more convincing. Also, the experiment can also vary the LLM used for checking to demonstrate the generalization.

**Limitations:**

yes

**Strengths And Weaknesses:**

**Strengths:**
+ The paper’s three-tier diagnostic framework is straightforward and easy to follow. Moving from simple perturbations to white-box attacks and then to closed-loop RL, it builds a layered picture of PRM vulnerabilities.

+ The rephrasing-based analysis of RL reward is novel. It is a clever way to separate stylistic exploitation from genuine reasoning improvement.

**Weaknesses:**
- The adversarial trigger experiments are built on a small number of trajectories it is hard to judge how generalizable the reported gains are. The proposed “basin volume” metric can be better explained to assess how meaningful the stability analysis is.

- There are also some experimental design choices that could be clarified further. The trigger insertion strategy differs across PRMs, which may confound cross-model comparisons, and the static perturbation pipeline depends heavily on GPT-4o for both generation and verification without additional human validation.

---

> ### Author Rebuttal · Authors · 2026-03-31
>
> Thank you for your valuable comments. We are glad you find the three-tier diagnostic framework straightforward and the rephrasing-based RL analysis novel. We address your questions below:
>
> ## R2-1 (W1/Q1): Stability of adversarial triggers on larger trajectory sets
>
> We designed the small-sample setup intentionally to show that minimal data suffices for attacks, but we agree that confirming stability on larger sets is important. To directly address this, we re-ran the adversarial token optimization on the **full** AIME24 dataset and evaluated on **all** AIME25 trajectories:
>
> [Table R2](https://image2url.com/r2/default/images/1774951086686-3c7c4d60-4510-458c-b339-5f0f5da6fbea.png)
>
> All three trends from the paper hold: Skywork-1.5B is highly vulnerable (Delta = +0.429 at k=100), Skywork-7B partially resists, and Qwen-7B resists exploitation. Peak rewards on Skywork-1.5B are lower than in the 8-trajectory setting (0.716 vs 0.954), which is expected: finding a single adversarial token across 30 diverse trajectories is harder than across 8. Results generalize to the full dataset.
>
> ## R2-2 (W2/Q2): Sensitivity to insertion position and trigger length
>
> We clarify that the insertion positions are not arbitrary but follow directly from how each model computes rewards. Skywork uses $R(\tau) = r_n$ (last step score), so adversarial tokens must be placed after the solution to influence the final step. Qwen uses $R(\tau) = \min\limits_i r_i$ (detecting the first wrong step), so tokens must be placed before the solution; otherwise they have no influence on the minimum (see footnote 1 in the paper). Placing tokens in the "wrong" position for either model would have no effect.
>
> To confirm this, we ran a position ablation for Skywork-1.5B with k=50 tokens:
>
> | Position | AIME24 (train) | AIME25 (+adv) | Delta |
> | --- | --- | --- | --- |
> | Start (before solution) | 0.409 | 0.187 | +0.051 |
> | Middle (within solution) | 0.298 | 0.129 | -0.006 |
> | End (after solution) | 0.487 | 0.274 | +0.138 |
>
> As shown, the end position is most effective, consistent with Skywork's last-step aggregation. Middle position has negligible effect (Delta = -0.006), confirming that position matters and our choices in the paper are well-motivated. The cross-model difference (Skywork vulnerable, Qwen resistant) stems from the aggregation objective: min-aggregation requires fooling every step simultaneously, which is inherently harder. We will add this ablation to the revised paper.
>
> ## R2-3 (W2/Q3): Validation beyond GPT-4o-based checking
>
> Our pipeline includes a manual review stage: for perturbation pairs with large reward deviations (|Delta R| > 0.5), we manually confirm the perturbation matches its intended category, identify generation artifacts, and filter out ambiguous cases. We have included these details in Appendix A.3. We agree with the reviewer that this is a crucial step, and keeping it in the appendix makes it easy to overlook, we will move these details to the main paper in the final version.
>
> To further strengthen this, we ran Claude Sonnet 4.6 as an independent equivalence checker on all perturbation samples:
>
> | Perturbation | GPT-4o Equiv | Sonnet 4.6 Equiv | Agreement |
> | --- | --- | --- | --- |
> | Rephrase | 95.1% | 96.9% | 95.7% |
> | Concise | 93.4% | 93.6% | 93.5% |
> | Verbose | 93.9% | 92.5% | 93.8% |
> | Incorrect Assumption | 26.2% | 7.5% | 80.0% |
>
> For semantics-preserving perturbations, both models consistently agree on equivalence at 93.5-95.7%. For semantics-altering perturbations (incorrect assumption), Sonnet 4.6 is stricter (7.5% equiv vs GPT-4o's 26.2%), which is the expected behavior for a stronger model. We also filtered to doubly-verified samples (both models confirm equivalence) and found PRM reward shifts nearly identical to single-LLM filtering (e.g., Skywork rephrase: mean Delta = -0.026 in both cases), confirming the original pipeline was reliable. We will add full details in the camera-ready version of the paper.

---

> > ### Author Rebuttal · Reviewer_u1Vo · 2026-04-03
> >
> > Thank the author for the detailed rebuttal. My concerns have been mostly resolved, and I will keep the positive score.

---

### Official Review · Reviewer_d6CE · 2026-03-12

**Soundness:** 3
**Presentation:** 3
**Significance:** 2
**Originality:** 2
**Overall Recommendation:** 4
**Confidence:** 3

**Summary:**

The paper provides a thorough analysis of the robustness of process reward models (PRM), including a static perturbation analysis, a gradient-based attack analysis, and a reward hacking analysis. The analysis showed several things that the PRM is not robust to.

**Compliance With Llm Reviewing Policy:**

Affirmed.

**Final Justification:**

The paper is technically solid with thorough experiments and evaluations, especially with the new results added in rebuttals. The paper's originality is one limitation. I personally slightly lean towards acceptance after the rebuttal.

**Key Questions For Authors:**

1. What is the difference between the static perturbation introduced compared to the rephrasing intervention in section 6.3? The paper seems to suggest that the reward model is quite robust to the static perturbation, yet not robust to the rephrasing intervention.
2. Can the authors release more details on the RL section, and conduct the experiments on more dataset+base model pairs?

**Limitations:**

yes

**Strengths And Weaknesses:**

**Strengths**

The paper is technically solid overall, and the writing is clear and easy to follow. It offers useful empirical insights into current PRMs, in particular suggesting that many existing PRMs behave more like fluency or plausibility checkers than true reasoning verifiers. I also found the central question important.

**Weakness**

1. My main concern is about how the PRM is used in the analysis, especially for the Skywork PRM. In my view, PRMs are valuable largely because they can provide denser step-level reward, helping address the sparse reward problem in RLVR. However, in the current analysis, it seems that for Skywork, the authors directly use the reward on the full trajectory rather than leveraging stepwise rewards directly. As a result, this setup evaluates the PRM more as a trajectory-level reward model than as a genuine dense process reward.
2. The RL analysis based on training Qwen-2.5-1.5B-Instruct on AIME is not very convincing. AIME is a very challenging benchmark, and for the small Qwen2.5-1.5B, the base performance is already extremely low. The current results may indeed suggest some degree of misalignment, but the results can be weak. I think the analysis would be stronger if it used either an easier benchmark or a more capable base model.

---

> ### Author Rebuttal · Authors · 2026-03-31
>
> Thank you for your valuable comments. We appreciate that you find the paper technically solid with clear writing and useful empirical insights. We address your questions below:
>
> ## R1-1: Trajectory-level vs. step-level PRM reward in RL
>
> We chose trajectory-level reward in our RL experiments because computing effective baselines for step-level PRM rewards remains an open challenge: trajectory-level GRPO uses the mean reward across rollouts as a natural baseline, but step-level credit assignment requires per-step baselines to reduce variance, and there is no established best practice for this.
>
> That said, we expect step-level PRM reward to be even more prone to reward hacking, since it directly optimizes each step's score against per-step biases revealed in Section 4. To confirm this, we ran three new experiments with Qwen2.5-7B-Instruct on AIME2024 using Skywork-1.5B PRM:
>
> - GRPO without PRM (correctness-only reward)
> - Trajectory-level GRPO with PRM (as in the paper)
> - Step-level GRPO with PRM, where the advantage at each step is computed with mean and standard deviation of all step rewards across rollouts for the same prompt. Since Skywork predicts the probability of correct completion, the mean reward across the rollout group serves as a natural per-step baseline.
>
> | GRPO variant | PRM Reward (start to current) | Accuracy (mean@8) |
> | --- | --- | --- |
> | GRPO w/o PRM | - | 12% to 63%, steadily climbing |
> | Trajectory-level | 0.36 to 0.88 | 11% to 20%, flat |
> | Step-level | 0.17 to 0.87 | 10% to 17%, flat |
>
> The GRPO w/o PRM reaches 63% accuracy with the same model and data, while both PRM variants plateau at \~17-20% despite PRM rewards climbing to \~0.88. This demonstrates that the policy learns to hack the PRM reward rather than improve reasoning. The contrast with the baseline confirms that reward-accuracy divergence is caused by PRM exploitation, not general RL instability.
>
> ## R1-2: Easier benchmark or more capable base model
>
> Our choice of a small model on a hard benchmark was deliberate: PRMs are most valuable when correctness-based reward provides little signal, i.e., when most rollouts are incorrect and outcome reward is sparse. This is precisely the regime where PRMs are deployed in practice to provide denser feedback, and the resulting optimization pressure makes reward hacking clearly observable.
>
> That said, we agree that demonstrating generality is important. In addition to the R1-1 experiments above (Qwen2.5-7B on AIME), we ran Qwen2.5-1.5B on the easier MATH-500 benchmark (\~500 problems) with Skywork-1.5B PRM. PRM reward climbs from 0.34 to 0.84 while accuracy plateaus at 62%. The GRPO w/o PRM baseline on the same setup achieves 81% accuracy. Combined with the AIME results (63% baseline vs 20% with PRM), reward hacking generalizes across both difficulty levels and model scales. The full set of all new experiments and their results are compiled in Table R1 (R1-4).
>
> ## R1-3: Static perturbation vs. rephrasing intervention
>
> The apparent contradiction is itself the key finding:
>
> **Section 4:** We rephrase naturally-written correct trajectories and find PRMs are robust: |Delta R| < 0.1 (Table 3). PRMs are invariant to random style variation on normal text.
>
> **Section 6.3:** We rephrase Trajectory-level GRPO with PRM-trained outputs and observe a significant drop (R = 0.641 to 0.472). These outputs have been optimized by RL to exploit PRM preferences, amplifying specific stylistic patterns that correlate with high scores.
>
> **Why the difference?** PRMs have subtle stylistic preferences invisible under random variation but exploitable under optimization pressure. RL discovers and amplifies the patterns the PRM favors. Rephrasing disrupts these optimized patterns, revealing that 43% of GRPO's reward gain comes from style. This is why our three-tiered framework is necessary: static analysis alone would miss this. We will expand on this discussion in the final version to make the distinction clearer.
>
> ## R1-4: More details and dataset+model pairs for RL
>
> We conducted 6 new PRM-based RL experiments and 2 no-PRM baselines (1,000 training steps each), systematically varying four axes: base model scale (Qwen2.5-1.5B,
> Qwen2.5-7B), benchmark difficulty (AIME, MATH-500), PRM scale (Skywork-1.5B, Skywork-7B), and reward granularity (trajectory-level, step-level). The full results are compiled in Table R1 below:
>
> [Table R1](https://image2url.com/r2/default/images/1774948627896-dc7f2240-482f-4961-ac66-a55b3bf400e1.png)
>
> Reward hacking persists across all settings: PRM reward climbs to 0.82–0.94 while accuracy stays flat or plateaus well below no-PRM baselines (which reach 63% on AIME and 81% on MATH-500 with steadily climbing accuracy). Step-level reward exacerbates hacking (17% vs 20–23% for trajectory-level), and scaling the PRM from 1.5B to 7B increases exploitable reward without improving accuracy. We will add full hyperparameters, learning curves, and rephrasing intervention analysis in the revised paper.

---

> > ### Author Rebuttal · Reviewer_d6CE · 2026-04-03
> >
> > I thank the authors for their detailed rebuttal. The rebuttal clears many of my misunderstandings and provides stronger results for the paper. I will thus increase my rating to 4. I will suggest that the authors discuss the difference between static perturbation and RL-induced reward hacking more clearly in the revised version.

---

### Decision · Program_Chairs · 2026-04-30

**Decision:**

Accept (regular)

**Comment:**

All reviewers highlight the clarity of presentation and the importance of the central question. The proposed three-tier framework provides a structured and intuitive way to analyze PRM vulnerabilities, progressing from controlled perturbations to adversarial attacks and closed-loop RL settings. The empirical findings (especially the identification of reward hacking and the fluency-logic dissociation) are considered insightful and valuable. The rebuttal further clarified several misunderstandings and strengthened confidence in the experimental setup, leading reviewers to maintain or increase their scores to positive.

**The main concerns raised by reviewers include:**

1. Experimental limitations: Several reviewers pointed out that some experiments rely on relatively small sample sizes (e.g., limited trajectories or AIME subsets), which raises questions about robustness and generalizability. There are also suggestions to include more PRM variants (e.g., generative PRMs) and larger-scale datasets.

2. Evaluation design choices: Questions were raised regarding the use of trajectory-level rewards for certain PRMs, the reliance on GPT-4o for perturbation generation and verification, and inconsistencies in adversarial trigger setups across models. While partially addressed in the rebuttal, these aspects could be further clarified and strengthened.

3. Analysis depth: Some components, such as the basin volume metric, adversarial trigger stability, and distinctions between different diagnostic stages, would benefit from clearer explanation and more comprehensive validation.

Despite these limitations, the consensus among reviewers is positive, with all reviewers recommending weak accept after the rebuttal. The paper provides meaningful empirical insights into an important emerging topic, and its diagnostic perspective is likely to be useful for future research on reward modeling and alignment. The identified weaknesses primarily relate to experimental breadth and clarity rather than fundamental flaws.